# Topological Relational Learning on Graphs

**Yuzhou Chen**
Department of Electrical Engineering
Princeton University
yc0774@princeton.edu

**Baris Coskunuzer**
Department of Mathematical Sciences
University of Texas at Dallas
coskunuz@utdallas.edu

**Yulia R. Gel**
Department of Mathematical Sciences
University of Texas at Dallas and
National Science Foundation
ygl@utdallas.edu

## Abstract

Graph neural networks (GNNs) have emerged as a powerful tool for graph classification and representation learning. However, GNNs tend to suffer from oversmoothing problems and are vulnerable to graph perturbations. To address these challenges, we propose a novel topological neural framework of topological relational inference (TRI) which allows for integrating higher-order graph information to GNNs and for systematically learning a local graph structure. The key idea is to rewire the original graph by using the persistent homology of the small neighborhoods of nodes and then to incorporate the extracted topological summaries as the side information into the local algorithm. As a result, the new framework enables us to harness both the conventional information on the graph structure and information on the graph higher order topological properties. We derive theoretical stability guarantees for the new local topological representation and discuss their implications on the graph algebraic connectivity. The experimental results on node classification tasks demonstrate that the new TRI-GNN outperforms all 14 state-of-the-art baselines on 6 out 7 graphs and exhibit higher robustness to perturbations, yielding up to 10% better performance under noisy scenarios.

## 1 Introduction

Node classification is one of the most active research areas in graph learning. The target here is, given a single attributed graph $\mathcal{G}$ and a small subset of nodes with prior label information, to predict labels of all remaining unlabelled nodes. Applications of node classification are very diverse, ranging from customer attrition analytics to tracking corruption-convictions among politicians. Graph neural networks (GNNs) offer a powerful machinery for addressing such problems on large heterogeneous networks, resulting in an emerging field of geometric deep learning (GDL) which adapts deep learning to non-Euclidean objects such as graphs [9, 51, 54, 16].

Although GDL achieves a highly competitive performance in various graph-based classification and prediction tasks, similarly to the image domain deep learning on graphs is often found to be vulnerable to graph perturbations and adversarial attacks [43, 50, 26]. In turn, most recent results [42, 19] suggest that local graph information may be invaluable for robustifying GDL against graph perturbations and adversarial attacks. Also, as shown by [36, 4, 53] in conjunction with network community learning, local algorithms (i.e., algorithms based only on a small radius neighborhood around the node) may demonstrate superior performance if coupled with side information in the form of a node labeling positively correlated with the true graph structure. Inspired by these results, the ultimate goal of this

35th Conference on Neural Information Processing Systems (NeurIPS 2021).

paper is to introduce the idea of local algorithms with *local topological side information* on similarity of node neighborhood shapes into GNN. By *shape* here, we broadly understand data characteristics which are invariant under continuous transformations such as stretching, bending, and compressing, and to study such graph shapes properties, we invoke the machinery of topological data analysis (TDA) [21, 11].

**Topological Relational Inference: from Matchmaking to Adversarial Graph Learning and Beyond** In particular, to capture more complex graph properties and enhance model robustness, we introduce the concept of topological relational inference (TRI) and propose two novel options for information passing which rely on the local topological structure of each node: (i) Inject a new topology-induced multiedge between two nodes based on shape similarity of their local neighborhoods; (ii) Enrich each individual node features by harnessing local topological side information from its neighbors. The rationale behind our idea is multi-fold. First, we assess not only global graph topology and relationships between the feature sets of two individual nodes, as with the current diffusion mechanisms and random walks on graphs, but we also examine important *interactions between shapes of the feature sets of the node neighborhoods*. As a result, our new topology-induced multigraph representation (TIMR) of graph $\mathcal{G}$ in (i) strengthens the relationship among nodes which might not be (yet) connected by a visible (or "tangible") edge in $\mathcal{G}$ but whose higher order intrinsic characteristics are very similar. The intuitive analogy here might be with a matchmaking agency who aims to connect two people on a blind date, based on a careful assessment of interplay among similarities of their interests, socio-demographics as well as that of their close friends, rather than bringing in two individuals together through a random walk among all available profiles of potential candidates. Another example is the customer churn and retention analytics based on peer effects [15]. That is, the goal here is to classify the node as potential churn customer based not only on the individual attributes but on interactions among its neighbor attributes. Second, such a new multiedge in (i), based on shape similarity of node local neighborhoods may assist not only the matchmaking agency in coupling the most appropriate individuals (or link prediction in less romantic and more technical terms), but to enhance resilience of the graph structure and associated graph learning. Indeed, a new multiedge in (i) can be used as an essential remedial ingredient in graph defense mechanisms as criteria to clean the perturbed graph, i.e., to reconstruct edges removed by the attacker or to suppress edges induced by the attacker, depending on the strength of the topology-induced link. Third, by learning shape properties within each local node neighborhood, our TRI allows for systematic recovering of intrinsic higher-order interactions among nodes well beyond a level of pair-wise connectivity, and such local topological side information in (ii) enhances accuracy in node classification tasks even on clean graphs. Significance of our contributions are the following:

- We propose a novel perspective to graph learning with GNN – topological relational inference, based on the idea of similarity among shapes of local node neighborhoods.

- We develop a new topology-induced multigraph representation of graphs which systematically accounts for the key local information on the attributed graphs and serves as an important remedial ingredient against graph perturbations and attacks.

- We derive theoretical stability guarantees for the new local topological representation of the graph and discuss their implications for the graph algebraic connectivity.

- Our expansive node classification experiments show that TRI-GNN outperforms 14 state-of-the-art baselines on 6 out 7 graphs and delivers substantially higher robustness (i.e., up to 10% in performance gains under noisy scenarios) than baselines on all 7 datasets.

## 2    Related Work

**Graph Neural Networks** Inspired by the success of the convolution mechanism on image-based tasks, GNNs continue to attract an increasing attention in the last few years. Based on the spectral graph theory, [10] introduced a graph-based convolution in Fourier domain. However, complexity of this model is very high since all Laplacian eigenvectors are needed. To tackle this problem, ChebNet [18] integrated spectral graph convolution with Chebyshev polynomials. Then, Graph Convolutional Networks (GCNs) of [30] simplified the graph convolution with a localized first-order approximation. More recently, there have been proposed various approaches based on accumulation of the graph information from a wider neighborhood, using diffusion aggregation and random walks. Such higher-order methods include approximate personalized propagation of neural predictions

(APPNP) [31], higher-order graph convolutional architectures (MixHop) [3], multi-scale graph convolution (N-GCN) [2], and Lévy Flights Graph Convolutional Networks (LFGCN) [14]. In addition to random walks, other recent approaches include GNNs on directed graphs (MotifNet) [35], graph convolutional networks with attention mechanism (GAT, SPAGAN) [48, 52], and graph Markov neural network (GMNN) [39]. Most recently, Liu et al. [34] consider utilizing information on the node neighbors' features in GNN, proposing Deep Adaptive Graph Neural Network (DAGNN). However, DAGNN does not account for the important information on the shapes of the node neighborhoods.

**Persistent Homology for Graph Learning** Machinery of TDA and persistent homology (PH) is increasingly widely used in conjunction with graph classification, that is, when the goal is to predict a label for the entire graph rather than for individual nodes. Such tools for graph classification with persistent topological signatures include kernel-based methods [46, 40, 33, 55, 32] and neural networks [24, 12]. All of the above methods consider the task of classifying graph labels and are based on assessing 'global' graph topology, while our focus is node classification and our approach is based on evaluating local topological graph properties (i.e., shape of individual node neighborhoods). Integration of PH to node classification is virtually unexplored. To the best of our knowledge, the closest result in this direction is PEGN-RC [56]. However, the key idea of PEGN-RC is distinctly different from our approach. PEGN-RC reweights **only each existing edge**, based on the topological information within its edge vicinity and, in contrast to TRI-GNN, neither compares any shapes, nor creates new or removes existing edges. Importantly, PEGN-RC does not integrate topology of both graph and node attributes, while TRI-GNN does.

# 3 Preliminaries on Topological Data Analysis and Persistent Homology

The machinery of topological data analysis (TDA) and, particularly, persistent homology offer a mathematically rigorous and systematic framework of tools to evaluate shape properties of the observed data, that is, intrinsic data characteristics which are invariant under continuous deformations such as stretching, compressing, and bending [58, 21, 37]. The main premise is that the observed data which can be, as in our case, a graph $\mathcal{G}$ or a point cloud in a Euclidean or any finite metric space constitute a discrete sample from some unknown metric space $\mathcal{M}$. Our goal is then to recover information on some essential properties of $\mathcal{M}$ which has been lost due to sampling. Persistent homology addresses this reconstruction task by counting occurrences of certain patterns, e.g., loops, holes, and cavities, within shape of $\mathcal{M}$. Such pattern counts and functions thereof, called *topological signatures* are then used to characterize intrinsic properties of $\mathcal{G}$.

The approach is implemented in two main steps. We start from associating $\mathcal{G}$ with some nested sequence of subgraphs $\mathcal{G}_1 \subseteq \mathcal{G}_2 \subseteq \ldots \subseteq \mathcal{G}_m = \mathcal{G}$. Then we monitor evolution of pattern occurrences (e.g., cycles, cavities, and more generally $n$-dimensional holes) in this nested sequence of subgraphs. To ensure a systematic and computationally efficient manner of pattern counting, we construct a simplicial complex $\mathcal{C}_i$ (e.g., a clique complex) induced by $\mathcal{G}_i$. The sequence $\{\mathcal{G}_i\}$ induces a *filtration*: nested sequence of simplicial complexes $\mathcal{C}_1 \subseteq \mathcal{C}_2 \subseteq \ldots \subseteq \mathcal{C}_m = \mathcal{C}$. Now we can not only track patterns but also evaluate the lifespan of each topological feature. Let $b_\sigma$ be the index of the simplicial complex $\mathcal{C}_{b_\sigma}$ at which we first record (i.e., birth) the $n$-dimensional topological feature $\sigma$ ($n$-cycle), while simplicial complex $\mathcal{C}_{d_\sigma}$ be the first complex we observe its disappearance (i.e., death). Then lifespan or *persistence* of the topological feature $\sigma$ is $d_\sigma - b_\sigma$. To evaluate all topological features together, we consider a *persistence diagram (PD)* where the multi-set $\mathcal{D}_n = \{(b_\sigma, d_\sigma) \in \mathbb{R}^2 : d_\sigma > b_\sigma\} \cup \Delta$ records the birth and deaths of all $n$-cycles in the filtration $\{\mathcal{C}_i\}$. Here, $\Delta = \{(t, t) | t \in \mathbb{R}\}$ is the diagonal set containing points in PD, counted with infinite multiplicity. Different persistent diagrams can be compared based on the cost of the optimal matching between points of the two diagrams, while avoiding topological noise near $\Delta$ [13].

Depending on the question at hand, we can construct different suitable filtrations relevant to the problem. In this paper, we consider two different filtrations. First, we consider the sublevel filtration based on a node degree function $f : \mathcal{V} \mapsto \mathbb{N}$. As degree is an integer valued function, so are our thresholds $\{\alpha_i\} \subset \mathbb{N}$. Our sublevel filtration is then defined as follows. Let $\mathcal{V}_{\alpha_i} = \{u \in \mathcal{V} \mid f(u) \leq \alpha_i\}$. Then, $\mathcal{G}_{\alpha_i}$ is the subgraph generated by $\mathcal{V}_{\alpha_i}$. In particular, the edge $e_{uv} \in \mathcal{E}$ is in $\mathcal{G}_{\alpha_i}$ if both $u$ and $v$ are in $\mathcal{V}_{\alpha_i}$. We call this *degree based filtration*. In addition, we consider a second filtration defined by the edge-weight function on the graph where edge weights are induced by similarity of node attributes. We call this *attribute based filtration* (See Section 4). Note that degree based filtration

is induced only by the graph properties, while attribute based filtration is constructed by using the features coming from the observed data.

# 4 Topological Relational Inference Graph Neural Network (TRI-GNN)

**Problem Statement**   Let $\mathcal{G} = (\mathcal{V}, \mathcal{E})$ be an attributed graph with a set of nodes $\mathcal{V}$, a set of edges $\mathcal{E} \subseteq \mathcal{V} \times \mathcal{V}$ and $e_{uv} \in \mathcal{E}$ denoting an edge between nodes $u, v \in \mathcal{V}$. The total number of nodes in $\mathcal{G}$ is $N = |\mathcal{V}|$. Let $W \in \mathbb{R}^{N \times N}$ be a $N \times N$-adjacency matrix of $\mathcal{G}$ such that $\omega_{uv} = 1$ for $e_{uv} \in \mathcal{E}$ and 0 otherwise, and $D$ be a diagonal $N \times N$-degree matrix with $D_{uu} = \sum_v \omega_{uv}$. For undirected graphs $W$ is symmetric (i.e., $W = W^\top$). To feed in directed graphs into the graph neural network architecture, we set $W' = (W^\top + W)/2$. Finally, each $u \in \mathcal{V}$ is equipped with a set of node features, i.e., $X_u = (X_{u1}, X_{u2}, \ldots, X_{uF})$ represents an $F$-dimensional feature vector for node $u \in \mathcal{V}$, and $X$ is a $N \times F$-matrix of all node features. Our objective is to develop a robust semi-supervised classifier such that we can predict unknown node labels in $\mathcal{G}$, given some training set of nodes in $\mathcal{G}$ with prior labeling information. Figure 1 illustrates our TRI-GNN model framework.

## 4.1 Topology-induced Multigraph Representation

The first step in our TRI model with the associated Topology-induced Multigraph Representation (TIMR) of $\mathcal{G}$ is to define topological similarity among two node neighborhoods. The key goal here is to go beyond the vanilla local optimization algorithms which capture only pairwise similarity of node features [36] and beyond only triangle motifs as descriptors of higher-order node interactions [35, 47]. Our aim is to systematically extract *all* $n$-dimensional topological features and their persistence in each node neighborhood and then to compare node neighborhoods in terms of their exhibited shapes.

**Definition 1** (Weighted $k$-hop Neighborhood).  An induced subgraph $\mathcal{G}_u^k = (\mathcal{V}_u^k, \mathcal{E}_u^k) \subseteq \mathcal{G}$, equipped with an edge-weight function $\tau$ induced by similarity of node features in $\mathcal{G}_u^k$, is called a *weighted k-hop neighborhood* of node $u \in \mathcal{V}$ if: (1) for any $v \in \mathcal{V}_u^k$, the shortest path between $u$ and $v$ is at most $k$; (2) edge-weight function $\tau : \mathcal{V}_u^k \times \mathcal{V}_u^k \mapsto \mathbb{R}_{\geq 0}$ is such that for any $v, w \in \mathcal{V}_u^k$ with $e_{vw} \in \mathcal{E}_u^k$, $\tau_{vw} = ||X_v - X_w||$, where $|| \cdot ||$ is either Euclidean distance (in the case of continuous node features), Hamming distance (in the case of categorical node features), Heterogeneous Value Difference Metric (HVDM) or other distance appropriate for mixed-type data [20]. If $\tau_{vw} \equiv 1$ for any $e_{vw} \in \mathcal{E}_u^k$, $\mathcal{G}_u^k$ reduces to a conventional $k$-hop neighborhood.

Armed with the edge-weight function $\tau$ induced by node attributes (see Definition 1) or with a node degree function, we now compute a sublevel filtration within each node neighborhood and track lifespan of each extracted topological feature, e.g., components, loops, cavities, and $n$-dimensional holes (see previous section). Here we consider two cases: attribute based filtration ($\mathcal{G}_u^k$ is equipped with an edge-weight function $\tau$ based on node attributes) and degree based filtration ($\mathcal{G}_u^k$ is an unweighted graph with $\tau \equiv 1$). We can then compare how topologically similar shapes exhibited by node neighborhoods in terms of Wasserstein distance among their persistence diagrams.

**Definition 2. (Topological Similarity among $k$-hop Neighborhoods)** Let $\mathcal{D}(\mathcal{G}_u^k)$ and $\mathcal{D}(\mathcal{G}_v^k)$ be persistence diagrams of the weighted $k$-hop neighborhood subgraphs $\mathcal{G}_u^k$ and $\mathcal{G}_v^k$ of nodes $u$ and $v$, respectively. We measure topological similarity between $\mathcal{G}_u^k$ and $\mathcal{G}_v^k$ with Wasserstein distance between the corresponding persistence diagrams as $d_{W_p}(\mathcal{G}_u^k, \mathcal{G}_v^k) = \inf_\gamma \left( \sum_{x \in \mathcal{D}(\mathcal{G}_u^k) \cup \Delta} ||x - \gamma(x)||_\infty^p \right)^{\frac{1}{p}}$, where $p \geq 1$ and $\gamma$ is taken over all bijective maps between $\mathcal{D}(\mathcal{G}_u^k) \cup \Delta$ and $\mathcal{D}(\mathcal{G}_v^k) \cup \Delta$, counting their multiplicities. In our analysis we use $p = 1$.

As noted by [36], integrating neighboring nodes whose labeling is positively correlated with the true cluster structure, as a side information to a community recovery process may lead to substantial performance gains. Inspired by these results, we not only inject topological side information into GNN but also distinguish this side information w.r.t. its strength. We propose a new topology-induced multigraph representation of $\mathcal{G}$, where a multiedge between nodes $u$ and $v$ comprises information on (tangible) connectivity between $u$ and $v$ (i.e, existence of $e_{uv}$) as well as on strong and weak topological similarity of the local neighborhoods of $u$ and $v$, irregardless whether $e_{uv}$ exists.

**Definition 3. (Topology-induced Multigraph Representation (TIMR))** Consider a graph $\mathcal{G} = (\mathcal{V}, \mathcal{E})$, with $d_{W,p}(\mathcal{G}_u^k, \mathcal{G}_v^k)$ as a measure of topological similarity among two $k$-hop neighborhoods

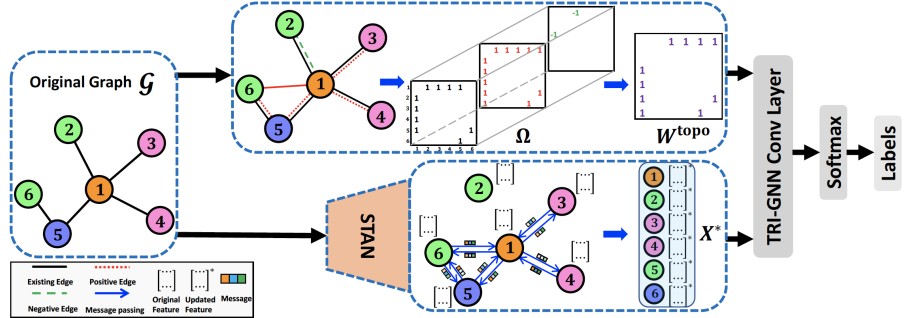

Figure 1: The TRI-GNN semi-supervised learning framework (for more details see Appendix B).

$\mathcal{G}_u^k$ and $\mathcal{G}_v^k$, for all $u, v \in \mathcal{V}$ (see Definition 1). Then a topology-induced multigraph representation of $\mathcal{G}$ is defined as $\Omega = (\mathcal{V}, \mathcal{E}^{\text{topo}})$, where $\mathcal{E}^{\text{topo}}$ is a set of multiedges such that for any $u, v \in \mathcal{V}$ and thresholds $\epsilon_1, \epsilon_2 \in \mathbb{R}^+$

$$e_{uv}^{\text{topo}} = \{ \mathbb{1}_{e_{uv} \in \mathcal{E}}, \mathbb{1}_{d_{W,p}(\mathcal{G}_u^k, \mathcal{G}_v^k) \in [0, \epsilon_1]}, -\mathbb{1}_{d_{W,p}(\mathcal{G}_u^k, \mathcal{G}_v^k) \in [\epsilon_2, \infty)} \}. \tag{1}$$

Note that $\{d_{W,p}(\mathcal{G}_u^k, \mathcal{G}_v^k) \in [0, \epsilon_1]\}$ and $\{d_{W,p}(\mathcal{G}_u^k, \mathcal{G}_v^k) \in [\epsilon_2, \infty)\}$ are incompatible events. Hence, multiedges have multiplicity at most 2 and $\mathcal{E}^{\text{topo}}$ is a multiset of $\{(1, 1, 0); (0, 1, 0); (1, 0, -1); (0, 0, -1)\}$. The intuition behind TIMR is to reflect a level of shape similarity among any two node neighborhoods, irregardless whether there exists an edge between these nodes in $\mathcal{G}$. Later, by using $\Omega$, we induce a graph $W^{topo}$ (Figure 1) defined as follows: If neighborhoods of two nodes in $\mathcal{G}$ are topologically similar, we add an edge between the nodes if there is none. If neighborhoods of two nodes in $\mathcal{G}$ are topologically dissimilar, we remove the edge between the nodes if one exists.

Alternatively, we can associate TIMR with *positive topology-induced* and *negative topology-induced* adjacency matrices, $W^{\text{topo}^+}$ and $W^{\text{topo}^-}$, respectively

$$W_{uv}^{\text{topo}^+} = \big[ 0 \leq d_{W,p}^f(\mathcal{G}_u^k, \mathcal{G}_v^k) < \epsilon_1 \big], \qquad W_{uv}^{\text{topo}^-} = \big[ \epsilon_2 < d_{W,p}^f(\mathcal{G}_u^k, \mathcal{G}_v^k) < \infty \big], \tag{2}$$

where $[\cdot]$ is an Iverson bracket, i.e., 1 whenever a condition in the bracket is satisfied, and 0 otherwise. Selection of hyperparameters $\epsilon_1$ and $\epsilon_2$ can be performed by assessing quantiles of the empirical distribution of shape similarities and then cross-validation. Thresholds $\epsilon_1$ and $\epsilon_2$ are used to reinforce meaningful edge structures and suppress noisy edges.

**How does TIMR help?** TIMR allows us not only to inject a new edge among two nodes if shapes of their multi-hop neighborhoods are sufficiently close, but also to eliminate an existing edge if topological distance among their multi-hop node neighborhoods is higher than predefined threshold $\epsilon_2$. That is, TIMR adds an edge between the nodes whose "similarity" is detected by persistent homology, and removes the edge between "topologically dissimilar" nodes. This way, in the new TIMR graph, similar nodes gets closer, and dissimilar nodes gets farther away, thereby assisting throughout the node classification process. As a result, TIMR also mitigates the impact of noise edges and reduces the effect of over-fitting. Furthermore, while we have not formally explored TIMR in conjunction with formal defense mechanisms against adversarial attacks, TIMR may be viewed as an essential remedial ingredient in defense. Indeed, TIMR offers an insight about one of the key defense challenges [27], namely, systematic criteria we should follow to clean the attacked graph – TIMR suggests to recover edges removed by the attacker with positive topology-induced links and to suppress edges induced by the attacker with negative topology-induced links.

## 4.2 TIMR Theoretical Stability Guarantees

We now establish theoretical stability properties of the TIMR average degree. The average degree is known to be closely related to performance in node classification tasks [28, 17, 1] and, hence, degree-dependent regularization is often used in adversarial training [38, 49, 44]. Here we prove that under perturbations of the observed graph $\mathcal{G}$, average degrees of the TIMR graphs of the original and distorted copies remain close. That is, the proposed TRI framework allows us to increase robustness

of the graph degree properties with respect to graph perturbations and attacks. Given that persistent homology representations are known to be robust against noise, our result appears to be intuitive. However, integration of graph persistent homology into node classification tasks and its theoretical guarantees remain yet an untapped area.

Let $\mathcal{G}^+$ and $\mathcal{G}^-$ be two graphs of the same order. Let $T^+$ and $T^-$ be TIMR graphs of $\mathcal{G}^+$ and $\mathcal{G}^-$, constructed by the degree based filtration. We define *local $k$-distance* between two graphs based on the Wasserstein distance $d_{W_p}$ between their persistence diagrams as follows. Let $u^{\pm}$ be a node in $\mathcal{G}^{\pm}$ and $\mathcal{G}_{u^{\pm}}^k$ be $k$-neighborhood of the $u^{\pm}$ in $\mathcal{G}^{\pm}$. Let

$$\mathbf{d}_k(u^+, u^-) = d_{W_1}(\mathcal{D}_0(\mathcal{G}_{u^+}^k), \mathcal{D}_0(\mathcal{G}_{u^-}^k)),$$

where $d_{W_1}$ is Wasserstein-1 distance (see Definition 1), and $\mathcal{D}_0(\mathcal{G})$ is the persistence diagram of 0-cycles. Let $\varphi : V^+ \to V^-$ be a bijection between node sets of $\mathcal{G}^+$ and $\mathcal{G}^-$, respectively. Then, the local $k$-distance between $\mathcal{G}^+$ and $\mathcal{G}^-$ is defined as

$$\mathfrak{D}^k(\mathcal{G}^+, \mathcal{G}^-) = \min_{\varphi} \sum_{u^+} \mathbf{d}_k(u^+, \varphi(u^+)).$$

We now have the following stability result:

**Theorem 1** (Stability of Average Degree). Let $\mathcal{G}^+$ and $\mathcal{G}^-$ be two graphs of same size and order. Let $T^{\pm}$ be the TIMR graph induced by $\mathcal{G}^{\pm}$, with thresholds $\epsilon_1$ and $\epsilon_2$. Let $\alpha^{\pm}$ be the average degree of $T^{\pm}$. Then, there exists a constant $K(\epsilon_1, \epsilon_2) > 0$ such that

$$|\alpha^+ - \alpha^-| \leq K(\epsilon_1, \epsilon_2)\mathfrak{D}^k(\mathcal{G}^+, \mathcal{G}^-).$$

The proof of the theorem is given in Appendix A in the supplementary material.

Furthermore, we conjecture the following result on stability of algebraic connectivity of TIMR graphs for the attribute based case. Here, the algebraic connectivity $\lambda_2(G)$ is the second smallest eigenvalue of the graph Laplacian $L(G)$, and it is considered to be a spectral measure to determine the robustness of the graph [45].

**Conjecture:** Let $\mathcal{G}$ be a graph, and let $\mathcal{G}'$ be a graph which is obtained by adding one edge $e$ to $\mathcal{G}$, i.e., $\mathcal{G}' = \mathcal{G} \cup e$. Let $T, T'$ be the TIMR graphs induced by $\mathcal{G}, \mathcal{G}'$, respectively. Then,

$$|\lambda_2(T') - \lambda_2(T)| \leq K(\epsilon_1, \epsilon_2)|\lambda_2(\mathcal{G}') - \lambda_2(\mathcal{G})|.$$

Note that this conjecture is not true for degree based TIMR graphs. The reason for that when one adds an edge $e_{uv}$ to $\mathcal{G}$, then this operation significantly changes $k$-neighborhoods of all nodes in $N_k(v_i)$ and $N_k(v_j)$. Since degree based TIMR construction depends on persistence diagrams of these $k$-neighborhoods, adding such an edge causes an uncontrollable effect on the algebraic connectivity. However, in attribute based construction, similarity is only based on the attributes, and the distances in the graph does not have an effect on edge addition/deletion decision. So, when the original graph $\mathcal{G}$ is perturbed by adding an edge, the attributes remain intact, and TIMR representations of the original and perturbed graphs are identical.

## 4.3 STAN: learning from Subgraphs, Topology and Attributes of Neighbors

For graphs with continuous or binary node features, higher-order form of message passing and aggregation (i.e., powers of the adjacency matrix) are shown to capture important structural graph properties that are inaccessible at the node-level. However, such higher-order architectures largely focus on global network topology and do not account for the important local graph structures. Also, performance of such higher-order graph convolution architectures tend to suffer from over-smoothing and be susceptible to noisy observations, since their propagation schemes recursively update each node's feature vector by aggregating over information delivered by all further nodes captured by a random walk.

Here we propose a new recursive feature propagation scheme **STAN** which updates the node features from **S**ubgraphs, **T**opology, and **A**ttributes of **N**eighbors. In particular, for each target node, STAN (i) converts these three features into a form of topological edge weights, (ii) calculates topological average of neighborhood features, (iii) aggregates them with the initial node features.

Let $X_u^{(0)} = X_u$ be the initial node features for $u \in \mathcal{V}$ and $\mathcal{T}$ be the number of STAN iterations. Then we iteratively update the feature representation of node $u$ using side information collected from nodes within its $k$-hop local neighborhood via $X_u^{(t+1)} = f_{\text{up}}\Big(\phi_{\text{aggr}}\big(X_u^{(t)}, \alpha(u) \sum_{v \in \mathcal{V}_k(u)} \hat{d}_{uv} \cdot X_v^{(t)}\big)\Big)$, where $f_{\text{up}}$ is the update function such as a *multi-layer perceptron (MLP)* and a *gated network*, $\phi_{\text{aggr}}$ is function that aggregates topological features into node features, such as *sum* and *mean*; $\alpha(u)$ is a weighting factor which can be set either as a hyperparameter or a fixed scalar, and $\hat{d}_{uv}$ are normalized topological edge weights

$$\hat{d}_{uv} = \frac{\exp\left[\left(d^f_{W,p}(\mathcal{G}_u, \mathcal{G}_v)\right)^{-1}\right]}{\sum_{v \in \mathcal{V}_k(u)} \exp\left[\left(d^f_{W,p}(\mathcal{G}_u, \mathcal{G}_v)\right)^{-1}\right]}.$$

The core principle of STAN is to assign different importance scores to nodes, depending on how similar topologically their neighborhoods are to the target node, and to control the impact on the target node prediction when aggregating neighborhoods' information. For instance, suppose target node is $u$ and we also have nodes $v$ and $w$. We assign a higher weight to the edge between $u$ and $v$ (if the shapes of their neighborhoods are more similar) and a lower weight to the edge between $u$ and $w$ (if neighborhoods of $u$ and $w$ have more distinct topology). As a result, the updated node features of $u$ will be more affected by $v$.

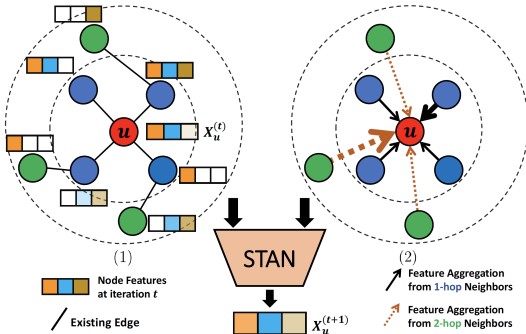

Figure 2: STAN for node feature vectors extension. The target node $u$ (red) with 2-hop neighborhood, where four 1-hop neighbors (blue) and three 2-hop neighbors (green). Each node is represented by a 3-component feature vector. We include more discussion on the STAN in Appendix B.

## 4.4 Convolution based TRI-GNN Layer

We now turn to construction of the TRI-GNN layer. Let $\boldsymbol{W} \in \mathbb{R}^{N \times N \times 3}$ be a 3-dimension tensor, where $N$ is the number of nodes and 3 is the number of candidate adjacency matrices (i.e., $W$, $W^{\text{topo}^+}$, and $W^{\text{topo}^-}$). Let $\boldsymbol{W}_{uvr}$ be the $r$-th type of edge between nodes $u$ and $v$ of $\boldsymbol{W}$, where $u \in \{1, \ldots, N\}$, $v \in \{1, \ldots, N\}$, and $r \in \{1, 2, 3\}$. That is, we denote $W$, $W^{\text{topo}^+}$, and $W^{\text{topo}^-}$ by setting $r = 1, 2, 3$. We then consider a *joint topology-induced* adjacency matrix $W^{\text{topo}}$

$$W_{uv}^{\text{topo}} = \begin{cases} 1, & \text{if } \sum_r \boldsymbol{W}_{uvr} > 0 \\ 0, & \text{otherwise} \end{cases}. \tag{3}$$

In the experiments, we utilize the classification function for the generalized semi-supervised learning as the default filter engine of TRI-GNN. Given $W^{\text{topo}}$, we compute topological distance-based graph Laplacian $L^{\text{topo}} = (\tilde{D}^{\text{topo}})^{-\sigma} \tilde{W}^{\text{topo}} (\tilde{D}^{\text{topo}})^{\sigma-1}$, where $\tilde{W}^{\text{topo}} = (W^{\text{topo}} + I)^{\rho}$, $\tilde{D}^{\text{topo}} = \sum_v \tilde{W}_{uv}^{\text{topo}}$, $\sigma \in [0, 1]$, and $\rho \in (0, \infty)$. Equipped with the new graph Laplacian $L^{\text{topo}}$ and node information matrix $X^{(\mathcal{T})}$ at iteration $\mathcal{T}$, TRI-GNN uses the following graph convolution layer

$$H^{(\ell+1)} = \psi\left(\mu\left(I - \mu L^{\text{topo}}\right)^{-1} H^{(\ell)} \Theta^{(\ell)}\right) = \psi\left(\mu \sum_{i=0}^{\infty} \left(\mu L^{\text{topo}}\right)^i H^{(\ell)} \Theta^{(\ell)}\right),$$

where $\mu \in (0, 1]$ is a regularization parameter, $H^{(\ell)} \in \mathbb{R}^{N \times F_\ell}$ and $H^{(\ell+1)} \in \mathbb{R}^{N \times F_{\ell+1}}$ are input and output activations for layer $\ell$ ($H^{(0)} = X^{(\mathcal{T})}$), $\Theta \in \mathbb{R}^{F_\ell \times F_{\ell+1}}$ are the trainable parameters of

TRI-GNN, and $\psi$ is an element-wise nonlinear activation function such as ReLU. In the experiments $\mu$ is selected from $\{0.1, 0.2, \ldots, 0.9\}$, and its choice impacts the number of terms in Taylor expansion. To reduce the computational complexity, based on the Taylor series expansion, we empirically find that $\max_i = \lceil \mathcal{R}\mu \rceil$ is a suitable rule of thumb (where $\mathcal{R} \in [2, 50]$), while the closer $\sigma$ is to 0, the more robust to the choice of regularization parameters the performance is. Note that in order to improve the robustness of TRI-GNN when it is exposed to noisy, we can consider convolution based TRI-GNN layer with parallel structure.

Table 1: Performance on semi-supervised classification. Average accuracy (%) and standard deviation (%) in ().

| Method | IEEE 118-Bus | ACTIVSg200 | ACTIVSg500 | ACTIVSg2000 | Cora-ML | CiteSeer | PubMed |
|--------|--------------|------------|------------|-------------|---------|----------|--------|
| ChebNet | 60.00 (3.11) | 80.63 (4.20) | 95.18 (1.00) | 80.04 (0.37) | 81.45 (1.21) | 70.23 (0.80) | 78.40 (1.10) |
| GCN | 52.86 (0.78) | 82.96 (1.66) | 90.33 (0.68) | 73.36 (0.41) | 81.50 (0.40) | 71.11 (0.72) | 79.00 (0.53) |
| MotifNet | 65.75 (0.77) | 73.20 (1.61) | 95.18 (0.53) | 82.00 (0.44) | - | - | - |
| ARMA | 70.55 (2.23) | 80.07 (3.30) | 94.33 (0.47) | 81.20 (0.23) | 82.80 (0.63) | 72.30 (0.44) | 78.80 (0.30) |
| GAT | 70.23 (1.80) | 83.56 (2.58) | 95.00 (0.47) | 83.00 (0.59) | 83.11 (0.70) | 70.85 (0.70) | 78.56 (0.31) |
| RGCN | 81.80 (1.79) | 83.35 (3.12) | 95.91 (0.40) | 85.23 (0.40) | 82.80 (0.60) | 72.13 (0.50) | 79.11 (0.30) |
| GMNN | 78.88 (2.50) | 84.13 (1.89) | 96.57 (0.52) | 86.21 (0.29) | 83.72 (0.90) | 73.10 (0.79) | **81.80 (0.53)** |
| LGCNs | 71.43 (2.20) | 83.78 (1.50) | 95.14 (0.45) | 85.57 (0.25) | 83.35 (0.51) | 73.08 (0.63) | 79.51 (0.22) |
| SPAGAN | 78.55 (2.25) | 81.83 (2.09) | 96.19 (0.40) | 86.00 (0.26) | 83.63 (0.55) | 73.02 (0.41) | 79.60 (0.40) |
| APPNP | 82.05 (2.24) | 81.21 (1.85) | 96.11 (0.45) | 86.77 (0.30) | 83.31 (0.53) | 72.30 (0.51) | 80.12 (0.20) |
| VPN | 82.19 (2.20) | 79.89 (1.82) | 96.23 (0.50) | 86.87 (0.33) | 81.89 (0.57) | 71.40 (0.32) | 79.60 (0.39) |
| LFGCN | 82.20 (2.30) | 81.11 (2.10) | 97.61 (0.47) | 87.74 (0.30) | 84.35 (0.57) | 71.89 (0.77) | 79.60 (0.55) |
| MixHop | 80.05 (2.50) | 80.53 (1.96) | 95.94 (0.40) | 86.10 (0.25) | 81.90 (0.81) | 71.41 (0.40) | 80.81 (0.58) |
| PEGN-RC | 82.30 (1.90) | 83.30 (1.10) | 97.20 (0.50) | 87.62 (0.74) | 82.70 (0.50) | 71.91 (0.63) | 79.40 (0.70) |
| TRI-GNN$_A$ | **82.80 (1.64)** | 84.93 (1.51) | 97.85 (0.56) | 88.82 (0.56) | 84.80 (0.55) | **73.45 (0.65)** | 79.80 (0.52) |
| TRI-GNN$_D$ | 82.67 (2.08) | **86.18 (0.20)** | **98.11 (0.43)** | **89.06 (0.44)** | **84.98 (0.49)** | 73.32 (0.48) | 79.70 (0.50) |

## 5 Experiments

We now empirically evaluate the effectiveness of our proposed method on seven node-classification benchmarks under semi-supervised setting with different graph size and feature type. We run all experiments for 50 times and report the average accuracy results and standard deviations.

### 5.1 Experimental Settings

**Datasets** We compare TRI-GNN with the state-of-the-art (SOA) baselines, using standard publicly available real and synthetic networks: (1) 3 citation networks [41]: Cora-ML, CiteSeer, and PubMed, where nodes are publications and edges are citations; (2) 4 synthetic power grid networks [8, 7, 23]: IEEE 118-bus system, ACTIVSg200 system, ACTIVSg500 system, and ACTIVSg2000 system, where each node represents a load bus, transformer, or generator and we use total line charging susceptance (BR_B) as edge weight. For more details see Table I (in Appendix C.1).

**Baselines** We compare TRI-GNN with 14 SOA GNNs: (i) 5 higher-order graph convolution architectures: LGCNs [22], APPNP [31], VPN [25], LFGCN [14], and MixHop [3]; (ii) 2 graph attention mechanism: GAT [48] and SPAGAN [52]; (iii) 4 GNNs with polynomial filters: ChebNet [18], GCN [30], MotifNet [35], and ARMA [6]; (iv) 2 GNNs by learning the generative distribution: GMNN [39] and RGCN [57]; and (v) 1 GNNs with topological information: PEGN-RC [56]. In the experiments, we implement two variants of TRI-GNN, i.e., TRI-GNN$_A$ (node attributes as filtration function) and TRI-GNN$_D$ (node degree as filtration function). In addition, in our robustness experiments, we select the 4 strongest baselines in each of the GNN areas: random walks, graph attention networks, higher-order graph convolutional models, and robust graph convolutional networks (i.e., APPNP, GAT, LFGCN, and RGCN). More details on experimental setup are in Appendix C.2. The source code of TRI-GNN is publicly available at `https://github.com/TRI-GNN/TRI-GNN.git`.

### 5.2 Performance on Node Classification

Table 1 shows the results for node classification results on graphs. We observe that TRI-GNN outperforms all baselines on CoraML, CiteSeer, and four power grid networks, while achieving comparable performance on PubMed. This consistently strong performance of TRI-GNN indicates utility of integrating more complex graph structures through aggregating higher-order topological

information. The best results are highlighted in bold for all datasets. As Table 1 shows, TRI-GNN$_D$ and TRI-GNN$_A$ outperform all baselines on all 7 but 1 benchmark datasets, yielding relative gains of 0.5-2.5% on clean graphs. Overall, TRI-GNN$_D$ appears to be the most competitive approach, especially for ACTIVSg200 and ACTIVSg500 systems. Nevertheless, TRI-GNN$_A$ tends to follow TRI-GNN$_D$ relatively closely. It validates that the framework of topological relational inference can integrate topological information across multigraph representation and feature propagation, respectively. On PubMed, GMNN achieves a better accuracy than both TRI-GNN versions due to PubMed exhibiting the weakest structural information with very few links per node on average, but both TRI-GNN models still deliver highly competitive results. In addition, we find that for sparser heterogeneous graphs, with lower numbers of node attributes per class, richer topological structure (i.e., synthetic ACTIVSg200 and ACTIVSg2000) and weaker dependency between attributes and classes, PH induced by the graph properties is more important than attribute-induced PH, while for denser, more homogeneous graphs as IEEE 118-bus, attribute-based filtration is the key. These insights echo findings in real citation networks.

**Ablation Study** Our TRI framework contains two key components: (i) encoding higher-order topological information via TIMR, (ii) STAN, i.e., updating the node features by aggregating information from subgraphs, topology, and neighborhood features. To glean a deeper insight into how different components help TRI-GNN to achieve highly competitive results, we conduct experiments by removing individual component separately, namely (1) TRI-GNN$_D$ w/o TIMR (i.e., denoted by $\Omega$), (2) TRI-GNN$_D$ w/o STAN, and (3) TRI-GNN$_D$ with TIMR and STN (where STN represents updating the node features by only aggregating information from subgraphs and topology without neighborhood features). As Table 2 shows, both TIMR $\Omega$ and STAN indeed yield significant performance improvements on all datasets. The ablation study on contribution of TIMR and STAN with TRI-GNN$_A$ are in Table II of Appendix C.3. The results show that all the components are indispensable.

Table 2: TRI-GNN$_D$ ablation study. Significance analysis is performed with one-sided two-sample $t$-test based on 50 runs; *, **, *** denote $p$-value $< 0.1, 0.05, 0.01$ (i.e., significant, statistically significant, highly statistically significant).

| Dataset | TRI-GNN$_D$ | w/o $\Omega$ | w/o STAN | with $\Omega$, STN |
|---|---|---|---|---|
| ACTIVSg200 | **86.18 (0.20)** | ***82.93 (1.32) | ***83.00 (3.17) | ***83.25 (1.58) |
| ACTIVSg500 | **98.11 (0.43)** | ***93.71 (1.03) | ***94.31 (1.20) | ***93.69 (1.50) |
| Cora-ML | **84.98 (0.49)** | ***84.26 (0.77) | ***84.00 (0.33) | ***84.40 (0.56) |
| CiteSeer | **73.32 (0.48)** | **73.11 (0.43) | 73.20 (0.55) | *73.19 (0.47) |

**Boundary Sensitivity** We select the $\epsilon_1$ and $\epsilon_2$ values based on quantiles of Wasserstein distances between persistence diagrams (see Figure 3a). The optimal choice of $\epsilon_1$ and $\epsilon_2$ can be obtained via cross-validation. For instance, the optimal quantile of $\epsilon_1$ and $\epsilon_2$ for ACTIVSg200 dataset is 0.55 and 2.50 respectively. We consider the lower and upper bounds for $\epsilon_1$ are 0.50 (minimum) and 0.05-quantile, respectively, and the lower and upper bounds for $\epsilon_2$ are 0.1-quantile and 6.74 (maximum), respectively. For $\epsilon_1$, we generate a sequence from 0.50 to 2.00 with increment of the sequence 0.05; for $\epsilon_2$, we generate a sequence from 2.50 to 6.74 with increment of the sequence 0.5. Based on cross-validation, we conduct experiments in different $\epsilon_1$ and $\epsilon_2$ combinations. Figures 3b and 3c show the performances with respect to different thresholds ($\epsilon_1$ and $\epsilon_2$) combinations (where $^\dagger$ denotes the $\epsilon_i$ (where $i \in \{1, 2\}$) is not fixed). From the results in Figure 3, we find that $(\epsilon_1, \epsilon_2)$ around $(0.55, 2.50)$ tend to deliver the optimal performance (accuracy (%)). The optimal results differ among graphs and depend on sparsity, label rates and graph higher-order properties. To better examine the boundary sensitivity, we further evaluate hyperparameters $\epsilon_1$ and $\epsilon_2$ on ACTIVSg500 in Appendix C.8.

**Robustness Analysis** To evaluate the model performance under graph perturbations, we test the robustness of the TRI-GNN framework under random attacks (here we present the results for TRI-GNN$_A$, and analogous robustness for TRI-GNN$_D$ are in Appendix C.4.). Random attack randomly generates fake edges and injects them into the existing graph structure. The ratio of the number of fake edges to the number of existing edges varies from 0 to 100%. As Figure 4 shows, TRI-GNN$_A$ consistently outperforms all baselines on both citation and power grid networks under random attacks, delivering gains up to 10% in highly noisy scenarios (see Cora-ML). These findings indicate that TRI-GNN$_A$ is again the most robust GNN node classifier under random attacks. By aggregating local

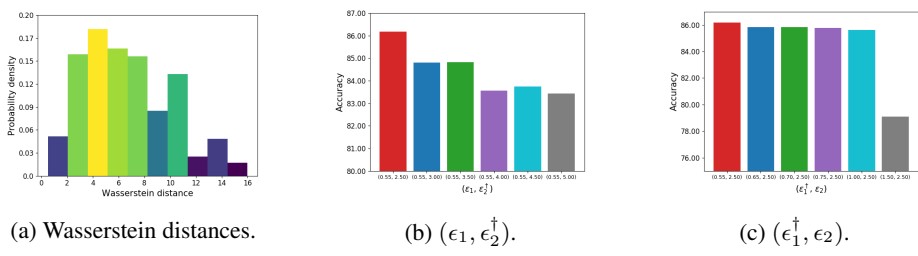

(a) Wasserstein distances.  (b) $(\epsilon_1, \epsilon_2^{\dagger})$.  (c) $(\epsilon_1^{\dagger}, \epsilon_2)$.

Figure 3: Hyperparameters $\epsilon_1$ and $\epsilon_2$ selection of TRI-GNN on ACTIVSg200 dataset.

topological information from node features and neighborhood substructures, TRI-GNN$_A$ can "absorb" noisy edges and, as a result, exhibits less sensitivity to graph perturbations and adversarial attacks.

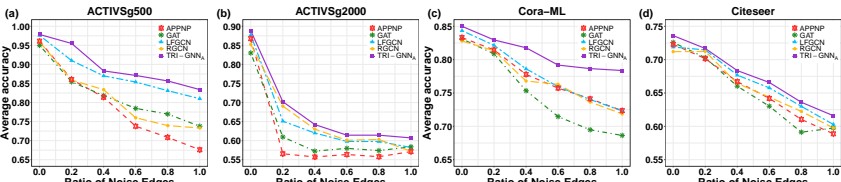

Figure 4: Node classification accuracy under random attacks.

**Computational Costs** In less extreme scenarios, the expected number of nodes in the $k$-hop neighborhood can be approximated by $\bar{d}^k$, where $\bar{d}$ is the mean degree of the graph and mathematical expectation is in terms of the node degree distribution. The computational complexity for Wasserstein distance between two persistence diagrams is $\mathcal{O}(M^3)$, where $M$ is the total number of barcodes in the both persistence diagrams. In the worst case scenario, if we use a very fine filtering function for persistence diagrams of $k$-hop neighborhoods, it would give at most $2\bar{d}^k$ barcodes (i.e., the number of barcodes in each $k$-hop neighborhood is at most $\bar{d}^k$ and we compare two $k$-hop neighborhoods). Considering we use the degree filtering function for our persistence diagrams, we expect to have much less ($\sim \bar{d}^k$) barcodes in the 0-th persistence diagram of a $k$-hop neighborhood by using the technique in [29]. Hence, the overall computational complexity would be of stochastic order $\mathcal{O}(N^2 \bar{d}^{\frac{3k}{2}})$. It may be costly to run PH for denser graphs (currently, computational complexity is the main roadblock for all PH-based methods on graphs) but as we noted earlier, for denser graphs we may use a degree filtration under lower $k$-hops which is still found to be very powerful even for small $k$. Table II in Appendix C.2 reports the average running time of PD generation and training time per epoch of our TRI-GNN model on all datasets.

## 6 Conclusion

We have proposed a novel perspective to node classification with GNN, based on the TDA concepts invoked within each local node neighborhood. The new TRI approach has shown to deliver highly competitive results on clean graphs and substantial improvements in robustness against graph perturbations. In the future, we plan to apply TRI-GNN as a part of structured graph pooling.

## Societal Impact and Limitations

Among the major limitations we currently see is probable existence of the bias of the proposed node classification approach due to potentially unbalanced population groups [5]. Indeed, in the context of social data analysis, given that we analyze peer neighborhoods which themselves tend to be largely formed by the same users as the target node and hence are likely to be dominated by the abundant group, such bias is very likely to exist and to be non-negligible. Mitigating it is a problem on its own but some ideas include using multiple filtration functions based on different node attributes so the node neighborhoods are made more diverse and are no longer dominated by a specific group or removing some potentially sensitive node attributes (e.g., gender) completely from the study (which has its own pros and cons in terms of accuracy).

## Acknowledgements

This work is sponsored by the National Science Foundation under award numbers ECCS 2039701, INTERN supplement for ECCS 1824716, DMS 1925346 and the Department of the Navy, Office of Naval Research under ONR award number N000142112226. Part of this material is also based upon work supported by (while serving at) the National Science Foundation. Any opinions, findings, and conclusions or recommendations expressed in this material are those of the author(s) and do not necessarily reflect the views of the National Science Foundation and/or the Office of Naval Research. The authors are also grateful to the NeurIPS anonymous reviewers for many insightful suggestions and engaging discussion which improved clarity of the manuscript and highlighted new open research directions.

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
