# Supplementary Material to "Topological Relational Learning on Graphs"

**Yuzhou Chen**
Department of Electrical Engineering
Princeton University
yc0774@princeton.edu

**Baris Coskunuzer**
Department of Mathematical Sciences
University of Texas at Dallas
coskunuz@utdallas.edu

**Yulia R. Gel**
Department of Mathematical Sciences
University of Texas at Dallas and
National Science Foundation
ygl@utdallas.edu

## A Proof of Stability of Average Degree

In this part, we prove the following stability result. We use the notation from the paper.

**Theorem 1** (Stability of Average Degree). Let $\mathcal{G}^+$ and $\mathcal{G}^-$ be two graphs of same size and order. Let $T^\pm$ be the TIMR graph induced by $\mathcal{G}^\pm$, with thresholds $\epsilon_1$ and $\epsilon_2$. Let $\alpha^\pm$ be the average degree of $T^\pm$. Then, there exists a constant $K(\epsilon_1, \epsilon_2) > 0$ such that

$$|\alpha^+ - \alpha^-| \leq K(\epsilon_1, \epsilon_2)\mathfrak{D}^k(\mathcal{G}^+, \mathcal{G}^-).$$

*Proof.* For simplicity, we assume only that we add edges to $\mathcal{G}^\pm$ to obtain $T^\pm$. The case for removing edges is similar. As $\mathcal{G}^+$ and $\mathcal{G}^-$ are of the same order, number of nodes $V(\mathcal{G}^\pm) = V(T^\pm)$, say $m$. Let $E(\mathcal{G})$ represents the number of edges in the graph $G$. Assume we added $k^\pm$ edges to $\mathcal{G}^\pm$ and get $T^\pm$, i.e. $E(T^\pm) - E(\mathcal{G}^\pm) = k^\pm$. Recall that average degree of a graph $\alpha(\mathcal{G}) = 2E(\mathcal{G})/V(\mathcal{G})$. Hence,

$$\alpha^+ = \alpha(T^+) = \frac{2E(T^+)}{m} = \frac{2(E(\mathcal{G}^+) + k^+)}{m}$$
$$= \alpha(\mathcal{G}^+) + \frac{2k^+}{m}.$$

Similarly, $\alpha^- = \alpha(T^-) = \alpha(\mathcal{G}^-) + \frac{2k^-}{m}$. As $\mathcal{G}^+$ and $\mathcal{G}^-$ are of the same size,

$$\left|\alpha^+ - \alpha^-\right| = \left|\frac{2k^+}{m} - \frac{2k^-}{m}\right| = \frac{2}{m}|k^+ - k^-|.$$

If we can bound this quantity in terms of $\mathfrak{D}^k(\mathcal{G}^+, \mathcal{G}^-)$, then the result follows. Notice that adding $k^+$ edges to $\mathcal{G}^+$ implies that there exist $k^+$ pairs of nodes $\{(u_1^+, v_1^+), (u_2^+, v_2^+), \ldots, (u_{k^+}^+, v_{k^+}^+)\}$ in $\mathcal{G}^+$ such that $\mathbf{d}_k(u_i^+, v_i^+) < \epsilon_1$. We claim that we can find suitable $K(\epsilon_1) > 0$ such that $k^+ \sim k^-$ delivering the desired inequality.

Assume $\mathfrak{D}^k(\mathcal{G}^+, \mathcal{G}^-) = \delta$ where $\delta > 0$. Let $\varphi : \mathcal{G}^+ \to \mathcal{G}^-$ be the best matching for this metric. Notice that

$$\mathbf{d}_k(\varphi(u^+), \varphi(v^+)) \leq \mathbf{d}_k(\varphi(u^+), u^+) + \mathbf{d}_k(u^+, v^+) + \mathbf{d}_k(v^+, \varphi(v^+))$$

35th Conference on Neural Information Processing Systems (NeurIPS 2021).

By assumption,

$$\mathbf{d}_k(u^+, \varphi(u^+)) \leq \mathfrak{D}^k(\mathcal{G}^+, \mathcal{G}^-) \;=\; \delta$$
$$\mathbf{d}_k(v^+, \varphi(v^+)) \leq \mathfrak{D}^k(\mathcal{G}^+, \mathcal{G}^-) \;=\; \delta,$$

which implies

$$\mathbf{d}_k(\varphi(u^+), \varphi(v^+)) \leq 2\delta + \epsilon_1. \tag{1}$$

Let $k^+(t)$ be the number of pairs of points in $\mathcal{G}^+$ with PD distance $< t$, and define $k^-(t)$ likewise. We compare these two functions with respect to $\delta$, PD distance of $\mathcal{G}^+$ and $\mathcal{G}^-$. Notice that both functions are monotone nondecreasing, and since there are only finitely many values, they are both locally constant. As $\varphi(u^+), \varphi(v^+)$ are both nodes in $\mathcal{G}^-$, the inequality (1) shows $k^+(t) < k^-(2\delta + t)$, i.e., for each pair $u^+, v^+ \in \mathcal{G}^+$ with $k$-neighborhoods of $u^+$ and $v^+$ are $t$-close, we will have the pair $\varphi(u^+), \varphi(v^+) \in \mathcal{G}^-$ whose $k$-neighborhoods are $(2\delta + t)$-close. Similarly, we have $k^-(t) < k^+(2\delta + t)$. By taking $t = s - 2\delta$, we have $k^+(s - 2\delta) < k^-(s)$. Hence, assuming $k^+(t) > k^-(t)$ at $t$, this implies $k^+(t) - k^-(t) < k^+(t) - k^+(t - 2\delta)$. Hence, bounding $\dfrac{k^+(t) - k^+(t - 2\delta)}{\delta}$ by a constant $K(t)$ would suffice to finish the proof as $\delta$ is the PD distance appearing in the right hand side of the desired inequality.

Now, let $r_0$ be the minimum distance between the threshold pairs $\{(b, d)\}$ on the persistence diagram grid. Then, $d_k(u, v) \geq r_0$ for any $u, v \in \mathcal{G}^\pm$. Let $M = m \cdot (m - 1)/2$ be the total number of pairs in $\mathcal{G}^\pm$, i.e. $k^\pm \leq M$. Let $K_0 = 2M/r_0$. Now note that if $2\delta < r_0$, $k^+(t) - k^-(t - 2\delta) = 0$ for any threshold $t$, as by assumption, both $k^\pm$ are locally constant for intervals of at least $r_0$ length. If $2\delta \geq r_0$, then $K_0 \cdot \delta \geq M$. As a result, since $k^+(t) - k^-(t) \leq M$ as $M$ is the total number of pairs in $\mathcal{G}^\pm$, the proof follows. $\qquad\square$

## B  More detailed information of TRI-GNN

Figure 1 provides the overview of framework of TRI-GNN. For Figure 2, we provide more detailed descriptions of STAN within the Figure caption.

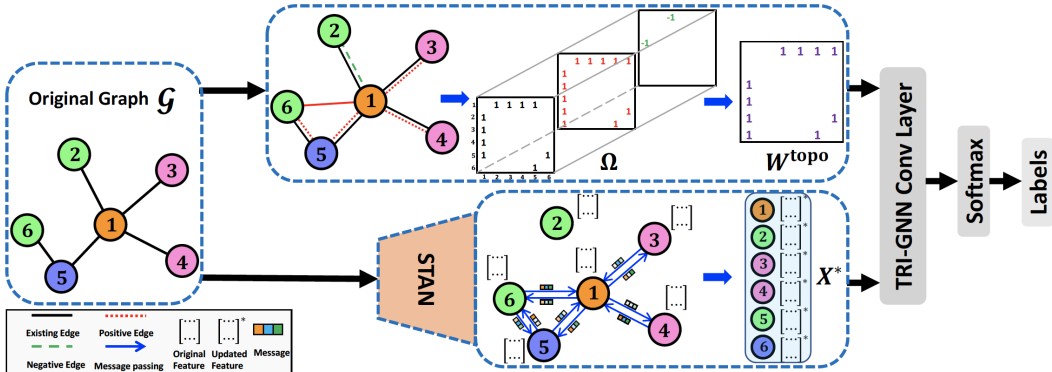

Figure 1: The framework of semi-supervised learning with TRI-GNN. Given the original graph $\mathcal{G}$, the upper part is the model architecture using topology-induced multigraph representation ($\Omega$) equipped with a set of multiedges (see Eq. (1) in Definition 3 of the main manuscript) to obtain the *joint topology-induced* adjacency matrix $W^{\text{topo}}$. Here **edge colored black** represents the original edge in the graph $\mathcal{G}$ (i.e., $W$), edge colored red represents *positive topology-induced* edge (i.e., $W^{\text{topo}^+}$), and edge colored green represents *negative topology-induced* edge (i.e., $W^{\text{topo}^-}$). The lower part is the model architecture using STAN to update node feature vectors (i.e., $X^*$) through weighted feature aggregation procedure based on topological distances. We then apply TRI-GNN convolutional layer and use the *joint topology-induced* adjacency matrix $W^{\text{topo}}$ and updated node feature vectors $X^*$ as input.

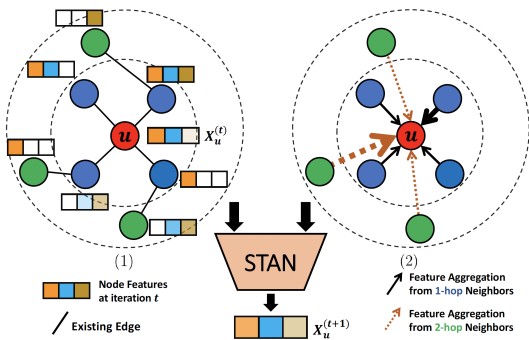

Figure 2: STAN for node feature vectors extension. The target node $u$ (red) with 2-hop neighborhood, where four 1-hop neighbors (blue) and three 2-hop neighbors (green). Each node is represented by a 3-component feature vector. STAN utilizes node features from 2-hop neighborhood and normalized reciprocal topological distances (i.e., $\hat{d}_{uv}$) between the target node and its 2-hop neighborhood to produce a new vector representation for $u$. (1) shows node feature vectors for all nodes in 2-hop neighborhood of $u$ at iteration $t$. (2) shows normalized reciprocal topological distances (which can be treated as edge weights) between 2-hop neighborhood and $u$, solid arrow indicates the weights during 1-hop neighborhood feature aggregation and dashed arrow indicates the weights during 2-hop neighborhood feature aggregation. After one iteration, STAN is applied to generate the feature vector for $u$ at iteration $t + 1$.

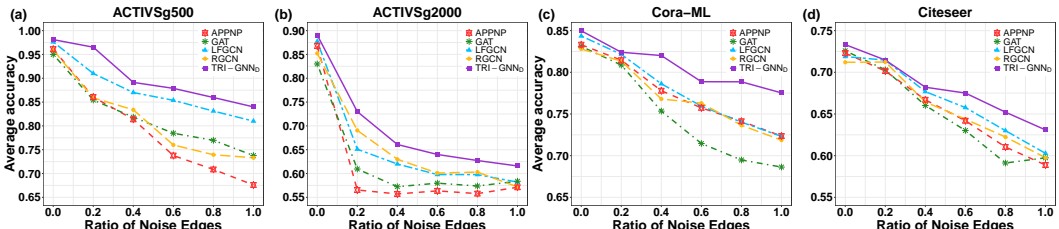

Figure 3: Node classification accuracy of TRI-GNN$_D$ under random attacks.

## C   More details on experiments

### C.1   Datasets

**Undirected networks** Cora-ML (this Cora dataset consists of Machine Learning papers), Citeseer and PubMed are three standard citation networks benchmark datasets used for semi-supervised learning evaluation [7]. In these citation networks, nodes represent publications, edges denote citation, the input feature matrix are bag of words and label matrix contain the class label of publication. We use the same data format as GCN, i.e., 20 labels per class in each citation network.

**Directed networks** We evaluate our method on four directed networks - IEEE 118-bus system (IEEE bus), ACTIVSg200, ACTIVSg500, and ACTIVSg2000 [2, 1, 5]. For IEEE 118-bus system, we consider a (unweighted-directed) graph as a model for the IEEE 118-bus system where nodes represent units such as generators, loads and buses, and edges represent the transmission lines. The input features of power grid network are generator active power upper bound (PMAX) and real power demand (PD) obtained from MATPOWER case struct. For ACTIVSg200, ACTIVSg500, and ACTIVSg2000, we also treat them as unweighted directed power grid networks. In particular, the input features are: (i) real power demand (PD); (ii) reactive power demand (QD); (iii) voltage magnitude (VM); (iv) voltage angle (VA); (v) base voltage (BASE_KV). For the power grid networks, we test them with $10\%$ label rate for training set, $20\%$ for validation and $70\%$ for test sets. Summary statistics of the data are summarized in Table I.

Table I: Dataset summary statistics.

| Dataset | Vertices | Edges | Features | Classes | Label rate |
|---|---|---|---|---|---|
| IEEE 118-Bus | 118 | 175 | 2 | 3 | 0.100 |
| ACTIVSg200 | 200 | 156 | 5 | 3 | 0.100 |
| ACTIVSg500 | 500 | 292 | 5 | 3 | 0.100 |
| ACTIVSg2000 | 2,000 | 1,917 | 5 | 3 | 0.100 |
| Cora-ML | 2,708 | 5,429 | 1,433 | 7 | 0.052 |
| CiteSeer | 3,327 | 4,732 | 3,703 | 6 | 0.036 |
| PubMed | 19,717 | 44,338 | 500 | 3 | 0.003 |

## C.2 Training Settings

For all experiments, we train our model utilizing Adam optimizer with learning rate $lr_1 = 0.01$ for undirected networks and $lr_2 = 0.1$ for directed networks. We use the same random seeds across all models (the experiments are repeated 5 times, each with the same random seed). To prevent over-fitting, we consider both adding dropout layer before two graph convolutional layers and kernel regularizers ($\ell_2$) in each layer. For undirected networks, we set the parameters of baselines by using two graph convolutional layers with 16 hidden units, $\ell_2$ regularization term with coefficient $5 \times 10^{-4}$, and dropout probability $p_{drop.}$ of 0.5 (for ARMA, except for number of hidden units, the hyperparameters setting are significantly different from others). For directed networks, we consider using a two-layer network but with $n_h^{\text{SOA}}$ hidden units (where $n_h^{\text{SOA}} \in \{16, 32, 64, 128, 256\}$), learning rate $lr_2$ with 0.001, 0.5 dropout rate and $\ell_2$ regularization weight of $5 \times 10^{-4}$ for baselines (for MotifNet, except for number of hidden units, the hyperparameters setting are significantly different from others). The best hyperparameter configurations, including dropout rate $p_{drop.}$, optimal number of hidden neurons $n_h^{\text{TRI-GNN}}$, regularization parameter $\mu$, the coefficient $\mathcal{R}$ (i.e., related to the number of power iteration steps), element-wise nonlinear activation function $\sigma(\cdot)$, of TRI-GNN for each dataset by using standard grid search mechanism (the optimal kernel regularization weight $\ell_2$ always equal to $5 \times 10^{-4}$). In STAN module, for the weighting factor $\alpha(u)$, we choose the best setting for each model, where $\alpha(u)$ is searched in $\{0.0, 0.1, 0.2, 0.3, 0.4, 0.5\}$; for the number of STAN iterations $\mathcal{T}$, we search $\mathcal{T}$ from $\{0, 1, 2\}$. We also use a parallel structure by stacking multiple convolution based TRI-GNN layers to attenuate the noise issue. For the choice of $k$ in the $k$-hop neighborhood, (i) for citation networks: we search $k$ from $\{1, 2, 3\}$ (where $k_{\text{Cora-ML}} = 2$, $k_{\text{CiteSeer}} = 1$, and $k_{\text{PubMed}} = 1$) and (ii) for synthetic power networks: we search $k$ from $\{3, 4, 5\}$ (where $k_{\text{IEEE}} = 5$, $k_{\text{ACTIVSg200}} = 5$, $k_{\text{ACTIVSg500}} = 3$, and $k_{\text{ACTIVSg2000}} = 5$). For threshold hyperparameters of topology-induced multiedge construction, in practice, we select the $\epsilon_1$ and $\epsilon_2$ values based on quantiles of Wasserstein distances between persistence diagrams. The optimal choice of $\epsilon_1$ and $\epsilon_2$ can be obtained via cross-validation. Specifically, we first search $\epsilon_1$ from $\{0.05, 0.1, 0.15, 0.2\}$-quantiles. With the best selction of $\epsilon_1$, we then search $\epsilon_2$ from $\{0.1, 0.15, \cdots, 0.95\}$-quantiles. Note that some most recent studies of [6, 3] have shown that dropping out edges helps reducing over-smoothing and improves training efficiency.

**Running Time** We report results for the average time (in seconds) taken for PD generation from $k$-hop neighborhood subgraph (in our experiment, $k \in \{1, 2, 3, 5\}$) and the average training time (in seconds) per epoch of TRI-GNN$_A$ for both undirected and directed networks on Tesla V100-SXM2-16GB.

## C.3 Ablation Study of TRI-GNN$_A$

The extended ablation study with statistical significance (i.e., $z$-test between TRI-GNN$_A$ and baselines) is in Table III. Except for the PubMed (the reason maybe that the PubMed is a very sparse network), in the power grids and Cora-ML, all TRI-GNN components are highly statistical significant, but in CiteSeer individual contributions of TIMR is statistical significant, while STAN is non-significant. This can be explained by CiteSeer highest sparsity and richest set of node attributes. Overall, TIMR is universally important.

Table II: Complexity of TRI-GNN$_A$: the average time (in seconds) to generate PD and training time per epoch.

| Dataset | PD generation | TRI-GNN$_A$ (per epoch) |
|---|---|---|
| IEEE 118-Bus | $5 \times 10^{-4}$s | 0.01s |
| ACTIVSg200 | $5 \times 10^{-4}$s | 0.01s |
| ACTIVSg500 | $1 \times 10^{-3}$s | 0.02s |
| ACTIVSg2000 | $1 \times 10^{-2}$s | 0.25s |
| Cora-ML | $7 \times 10^{-2}$s | 0.05s |
| CiteSeer | $1 \times 10^{-2}$s | 0.35s |
| PubMed | $2 \times 10^{-1}$s | 0.30s |

Table III: Ablation study of TRI-GNN$_A$ in accuracy (%) and standard deviation (%) in (); *, **, *** denote $p$-value $< 0.1, 0.05, 0.01$ (i.e., significant, statistically significant, highly statistically significant).

| Dataset | TRI-GNN$_A$ w/o $\Omega$ | TRI-GNN$_A$ w/o STAN | TRI-GNN$_A$ with $\Omega$, STN |
|---|---|---|---|
| IEEE 118-Bus | *82.20 (1.68) | *82.38 (2.00) | *82.21 (1.88) |
| ACTIVSg200 | ***82.75 (2.09) | **84.56 (1.75) | ***82.80 (2.73) |
| ACTIVSg500 | *97.85 (0.56) | *97.69 (0.63) | *97.54 (0.72) |
| ACTIVSg2000 | *88.59 (0.60) | *88.82 (0.56) | *88.63 (0.65) |
| Cora-ML | ***84.33 (0.63) | ***84.63 (0.61) | ***84.42 (0.70) |
| CiteSeer | 73.25 (0.70) | *73.10 (0.70) | *73.10 (0.63) |
| PubMed | 79.77 (0.51) | 79.71 (0.50) | 79.68 (0.63) |

## C.4 Random Attacks

Here we present the results for TRI-GNN$_D$ under random attacks. As demonstrated in Figure 3, we can see that the TRI-GNN$_D$ architecture indeed is capable to capture much richer local and global higher-order graph information. Particularly on Cora-ML, the performance of the 4 baselines drop rapidly as the ratio of perturbed edges increases, while TRI-GNN$_D$ successfully defends against the perturbations. Overall, the robustness of TRI-GNN$_D$ and TRI-GNN$_A$ (see Figure 4 in the main manuscript) are similar. These results indicate that the new TRI-GNN architecture with both types of the considered filtration functions is a highly competitive alternative for graph learning under adversarial perturbations.

## C.5 Diverse Set of TRI-GNN

We experimented shorter training sets and numbers of layers (see Table IV). TRI-GNN sustains its competitiveness, yielding one of the best results even under 20% reduction of the training set (compare Table 1 in the main manuscript). Higher order holes have not brought up any noticeable improvement as higher order homology are almost not met in relatively small node neighborhoods.

Table IV: TRI-GNN$_D$ with different training set sizes and number of layers. Number of runs is 50.

| Cora-ML | Training set size | | | |
|---|---|---|---|---|
| | 100% Train | 90% Train | 80% Train | 50% Train |
| Accuracy (%) (std) | **84.98 (0.49)** | 84.15 (0.55) | 84.00 (0.64) | 73.20 (0.50) |

| Cora-ML | Layers | | | |
|---|---|---|---|---|
| | 2 layers | 8 layers | 16 layers | 32 layers |
| Accuracy (%) (std) | **84.98 (0.49)** | 80.70 (0.75) | 78.06 (1.07) | 72.87 (1.30) |

## C.6 Comparison with GCN-LPA, NodeNet, and DFNet-ATT

Tables V and VI compare the performance of TRI-GNN$_D$ to other state-of-the-art graph classification baselines, i.e., DFNet-ATT [9], GCN-LPA [8], and NodeNet [4]. On Cora-ML$^\dagger$, we follow the settings of NodeNet, i.e., we split the Cora-ML dataset into training (80%) and test sets (20%). We can observe that TRI-GNN significantly outperforms GCN-LPA, NodeNet, and DFNet-ATT on these datasets.

Table V: Average accuracy (%) and standard deviation (%) in () on Cora-ML, ACTIVSg200, and ACTIVSg500.

| Method | TRI-GNN$_D$ | NodeNet |
|---|---|---|
| Cora-ML$^\dagger$ | **85.85 (0.71)** | 84.03 (0.40) |
| ACTIVSg200 | **86.18 (0.20)** | 80.15 (0.91) |
| ACTIVSg500 | **98.11 (0.43)** | 95.00 (0.50) |

Table VI: Average accuracy (%) and standard deviation (%) in () on Cora-ML and CiteSeer.

| Method | TRI-GNN$_D$ | GCN-LPA | DFNet-ATT |
|---|---|---|---|
| Cora-ML | **84.98 (0.49)** | 81.68 (0.93) | 84.65 (0.50) |
| CiteSeer | **73.32 (0.48)** | 71.40 (0.60) | 70.18 (0.71) |

## C.7 Additional Results

We also evaluate our methods on two relatively large datasets, i.e., ogbn-products and ogbn-arxiv. Table VII below shows that our proposed TRI-GNN always outperforms state-of-the-art methods in terms of classification accuracy on large networks. The results also indicate that our proposed model is capable to achieve highly promising results on very large networks.

Table VII: Average accuracy (%) and standard deviation (%) in () on ogbn-products and ogbn-arxiv.

| Method | TRI-GNN$_D$ | GCN | GAT | RGCN | APPNP |
|---|---|---|---|---|---|
| ogbn-products | **84.00 (0.007)** | 78.87 (0.003) | 80.02 (0.001) | 81.35 (0.005) | 83.17 (0.006) |
| ogbn-arxiv | **74.30 (0.003)** | 72.18 (0.002) | 72.47 (0.002) | 72.97 (0.001) | 72.10 (0.003) |

## C.8 Boundary Sensitivity on ACTIVSg500

We also evaluate the boundary sensitivity of hyperparameters $\epsilon_1$ and $\epsilon_2$ on ACTIVSg500 dataset. The results are presented in Figure 4. We can observe that, compared with $\epsilon_2$ (i.e., removing existing edges based on topological similarity among two node neighborhoods), TRI-GNN is more sensitive to $\epsilon_1$ (i.e., adding edges If two nodes in $\mathcal{G}$ are topologically similar).

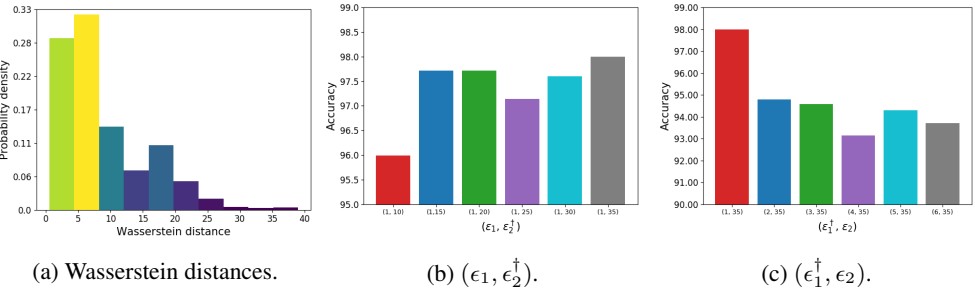

(a) Wasserstein distances.  (b) $(\epsilon_1, \epsilon_2^\dagger)$.  (c) $(\epsilon_1^\dagger, \epsilon_2)$.

Figure 4: Hyperparameters $\epsilon_1$ and $\epsilon_2$ selection of TRI-GNN on ACTIVSg500 dataset.