# OpenReview forum: "Topological Relational Learning on Graphs"
_NeurIPS.cc/2021/Conference — NeurIPS 2021 Poster_

### Official Review · Reviewer_uhB4 · 2021-07-12

**Rating:** 7
**Confidence:** 4

**Summary:**

The authors propose a way of rewiring an input graph using persistent homology and propose a class of GNNs that operate on top of this rewiring. The authors discuss and analyse the implications of such an approach for adversarial robustness, stability to graph perturbations and improving node classification performance.

**Ethical Concerns:**

EDIT: I have edited my response to flag the need for ethical review based on the motivating applications discussed by the authors, which might raise certain ethical issues.

**Ethics Review Area:**

["Inappropriate Potential Applications & Impact  (e.g., human rights concerns)"]

**Limitations And Societal Impact:**

The authors have not included a section on limitations and societal impact.

**Main Review:**

## Strong points

- To the best of my knowledge, graph rewiring via persistent homology (PH) is a new and original idea. So far, applications of persistent homology in geometric deep learning have focused primarily on graph classification, with the notable exception of the paper *Persistence Enhanced Graph Neural Network*, which the authors also comment on.
- I appreciate that the authors attempted to provide some theoretical stability guarantees about their proposed rewiring scheme. While PH has been used extensively in the field, its theoretical properties are rarely extended to the newly proposed methods. Therefore, this is an appreciable effort. At the same time, I have certain questions/doubts about the proof (see below).
- I appreciate the authors have added an ablation study, which clearly identifies the source of the improvements.
- The experiments have been run over a significant number of seeds (50), use a significant number of baselines, and the statistical significance is often explicitly stated.
- The robustness analysis nicely complements the theoretical results on stability to perturbations.
- Figure 1 does a good job of illustrating how the different pieces are coming together.

## Improvements

- I believe the main limitation of the work is that the experimental evaluation focuses only on a set of datasets with very small graphs. While these datasets could also provide valuable information about a method, it only shows how a method performs in a very limited setting. The limitations of these datasets have been widely studied recently (e.g. https://arxiv.org/abs/2003.00982). Therefore, I wished the authors complemented these experiments with (some) benchmarks from OGB (https://ogb.stanford.edu/) or the Benchmarking GNNs paper (https://arxiv.org/abs/2003.00982). These limitations also transfer over to the perturbation stability experiments.
- The presentation of the paper can be improved from many points of view:
  - The introduction to persistent homology is probably too abrupt for the average reader, who is likely not familiarised with persistent homology. For instance, the authors mention a simplicial complex when introducing PH, but simplicial complexes are never defined. Furthermore, I believe they are not even needed since this work is only interested in graphs (i.e. 1-dimensional simplicial complexes). At the same time, the authors mention k-dimensional topological features. But again, since we are talking about graphs we are not interested in k > 1. So these terms can be completely excluded to avoid confusing people and maintain only a minimally required technical jargon. A recent brief introduction to PH that I liked can be found in the background section of https://arxiv.org/abs/2102.07835. Notice how the authors use various examples to build an intuition. Perhaps the authors could borrow some ideas from there in the way the background material can be presented.
  - On the latter point, many definitions lack an intuitive explanation or justification. For instance, the intuition behind the Wasserstein distance is never explained. Or the intuition/motivation behind the expression for $W_{uv}^{topo}$ below line 264, which otherwise might seem a bit arbitrary.
  - While the Figures offer a pretty good visual description the captions are too brief and offer little information. For instance, the size of the arrows in Figure 2 and the colours of different elements are not explained at all.
  -  The paper is overflown with acronyms (TIMR, STAN, TRI-GNN) and the authors do not explain very well how all these pieces are coming together. I've understood this only after noticing in Figure 1. However, the main text never provides a unifying view and the sections seem a bit disconnected because of this. The order they are presented in could also be improved. I was very confused by how STAN and TRI-GNN interact since both of them produce multiple layers of features. This is roughly clear from the Figure, but still, I feel like I am missing many details.
 - I do not think it was explained (apologies if I missed it), but how are the results for the baselines produced? Are they taken from other papers or have the authors trained them? If it is the latter, how where these baselines fine-tuned to ensure a fair comparison.
- I am confused by lines 12-15 of the proof, which seem very hand-wavy to me. Could you explain more rigorously what is going on in those lines? I would have expected to see a series of inequalities leading to the desired inequality but this flow is quickly interrupted in line 12 where everything is just stated in words and the logical argument becomes very ambiguous.
- I believe the computational complexity of the method was not stated.
- Nit: Another topological approach for higher-order interactions which is not cited was introduced in https://arxiv.org/abs/2103.03212.

EDIT: After discussion, I decided to raise my score from 6 to 7.

**Needs Ethics Review:**

Yes

**Time Spent Reviewing:**

4

---

> ### Author Response · Authors · 2021-08-06
> **Response to Technical Questions (2/2: Q7, Q8, Q9)**
>
>
> Q7: "I am confused by lines 12-15 of the proof, which seem very hand-wavy to me. Could you explain more rigorously what is going on in those lines? I would have expected to see a series of inequalities leading to the desired inequality but this flow is quickly interrupted in line 12 where everything is just stated in words and the logical argument becomes very ambiguous."
>
> ${\bf A}$: We tried to be not too technical, that is why we explained this part in words. Here is how the proof proceeds. Since the inequalities are symmetric (matching $\varphi$ works both ways) without loss of generality, assume $k^+ > k^-$ . This means $k^+$ pair of points in $G^+$, have PD distance $<\epsilon_1$, and $k^-$ pair of points in $G^-$ have PD distance  $<\epsilon_1$. The inequalities show that at least $k^+$ pair of points in $G^-$ have PD distance $2\delta +\epsilon_1$. Then, for each given $\epsilon_1$, $k^+ - k^-$ is controlled by $\delta$ (there exists $K(\epsilon_1))$, the PD distance between $G^+$ and $G^-$. Since the difference of average degrees for same order graphs is given by $(k^+ - k^-)/m$, the proof follows.
>
> A bit more technical version would be as follows: If we define $k^+(t)$ is the number of pairs of points in $G^+$ with PD distance $< t$, and define $k^-(t)$ likewise, we compare these two functions with respect to $\delta$, PD distance of $G^+$ and $G^-$. Notice that both functions are monotone nondecreasing, and since there are only finitely many values, they are both locally constant.
>
> The inequality (Line 11-12) shows $k^+(t)<k^-(2\delta+t)$ and $k^-(t)<k^+(2\delta+t)$. By taking $t=s-2\delta$ in the , we have $k^+(s-2\delta)<k^-(s)$. Hence, assuming $k^+(t)>k^-(t)$ at $t$, this implies $k^+(t)-k^-(t)<k^+(t)-k^+(t-2\delta)$. Hence, bounding $\frac{k^+(t)-k^+(t-2\delta)}{\delta}$ by a constant $K(t)$ would suffice to finish the proof as $\delta$ is the PD distance appearing in the right hand side of the desired inequality.
>
> As we are studying finite graphs, we can do even better by finding a universal constant $K_0$ independent of $t$ as follows. Let $r_0$ be the minimum distance between the threshold pairs $\{(\alpha_i,\alpha_j)\}$ on the persistence diagram grid. Then, $d_k(u,v)\geq r_0$ for any $u,v \in G^\pm$. Let $M=m\cdot (m-1)/2$ be the total number of pairs $\{(u,v)\}$ in $G^\pm$, i.e. $k^\pm\leq M$. Let $K_0=2M/r_0$. Then if $2\delta<r_0$, $k^+(t)-k^+(t-2\delta)=0$ for any threshold $t$, as by assumption, both $k^\pm$ are locally constant for intervals of at least $r_0$ length. If $2\delta\geq r_0$, then $K_0\cdot \delta \geq M$. Then, since $k^+(t)-k^-(t)\leq M$ as $M$ is the total number of pairs in $G^\pm$, the proof follows.
>
> If requested, we could add this more technical version of the proof in the revision.
>
> Q8: I believe the computational complexity of the method was not stated.
>
> ${\bf A}$: We have reported the computational complexity (running time) about the average time to generate persistence diagram (PD) and training time (per epoch) in the supplementary material (Section C.5; line # 77 - 78). We also copy the table here. From the Table, we can see that persistence homology can be calculated efficiently for lower dimensional features with the complexity $O(m\log()m)$ for a graph with $m$ sorted edges.
> ++++++++++++++++++++++++++++++++++++++++++
>
> Dataset           |          PD generation         | TRI-GNN$_{\text{A}}$ (per epoch)
>
> ++++++++++++++++++++++++++++++++++++++++++
>
> IEEE 118-Bus  |   $5 \times 10^{-4} s$    |   $1 \times 10^{-2} s$
> ++++++++++++++++++++++++++++++++++++++++++
>
> ACTIVSg200    |   $5 \times 10^{-4} s$    |   $1 \times 10^{-2} s$
>
> ++++++++++++++++++++++++++++++++++++++++++
>
> ACTIVSg500    |   $1 \times 10^{-3} s$    |   $2 \times 10^{-2} s$
>
> ++++++++++++++++++++++++++++++++++++++++++
>
> ACTIVSg2000  |   $1 \times 10^{-2} s$    |   $2.5 \times 10^{-1} s$
>
> ++++++++++++++++++++++++++++++++++++++++++
>
> Cora-ML      $\quad $     |   $7 \times 10^{-2} s$    |   $5 \times 10^{-2} s$
>
> ++++++++++++++++++++++++++++++++++++++++++
>
> Citeseer      $\quad $     |   $1 \times 10^{-2} s$    |   $3.5 \times 10^{-1} s$
>
> ++++++++++++++++++++++++++++++++++++++++++
>
> PubMed       $\quad $   |   $2 \times 10^{-1} s$    |   $3 \times 10^{-1} s$
>
> ++++++++++++++++++++++++++++++++++++++++++
>
> Q9: Another topological approach for higher-order interactions which is not cited was introduced in https://arxiv.org/abs/2103.03212.
>
> ${\bf A}$:  Thank you for suggesting this paper! We got familiar with this paper (and its code) only after the submission deadline. We’ll certainly add it to references and agree that simplicial nets is probably the next generation of Geometric Deep Learning, though we cannot comment yet on how to potentially integrate it in an efficient manner for node classification.

---

> > ### Comment · Reviewer_uhB4 · 2021-08-18
> > **Response to Authors**
> >
> > Thank you for your detailed response! I am satisfied with most answers and the changes the authors are planning to make. Please find some additional comments below.
> >
> > __OGB Experiments__
> >
> > Thank you for running the OGB experiments! The results look solid and are close to the SOTA methods from the OGB leaderboards.
> >
> > I also appreciate the additional perturbation experiment on ogbn-products. However, there it might be more appropriate to also include the relative drop in performance for each method with respect to the non-perturbed setting. Since your method starts from an initial score that is much higher than the baselines, the fact that your method ends up with a higher absolute score at different perturbation levels does not necessarily mean it is more robust. In fact, percentage-wise, on a first look, it seems that the performance of the proposed method degrades slightly more than some other baselines.
> >
> > __"but for k=1 and higher, one needs to take at least k+1 dimensional clique complexes"__
> >
> > Persistence homology is still well-defined for k=1 without considering the 2-clique complex. In that case, it should capture information about all the cycles in the graph.
> >
> > __Computational complexity__
> >
> > The authors have only included the running time in the paper, not the computational complexity. The complexity of $O(m log(m))$ is only for computing persistence homology in dimension zero. The complexity increases for $k > 0$ and I believe the authors use such values. Furthermore, the method performs lots of additional computations besides computing the persistent homology. For instance, it computes the Wasserstein distance between persistence diagrams. What is the complexity associated with that? What is the complexity associated with the computation of $W^{topo}$? Finally, I would like to clearly see the computational complexity of all these individual components and the resulting overall complexity. This analysis should also take into account various hyper-parameters that might affect the complexity (like the dimension of the PD).
> >
> > __New proof__
> >
> > Thank you for adding a more detailed and technical proof! I did not have time to check it in detail and I will get back to the authors once I have done so.
> >
> > __Ethical issues__
> >
> > First of all, apologies for not mentioning this in my first response! I have made a note of it while reading the paper but somehow I forgot to transcribe it in my response here. However, I noticed that other reviewers have rightly flagged some possible ethical issues around the examples used in the paper.
> >
> > In my opinion, the examples chosen by the authors are not entirely appropriate and touch upon some very sensitive subjects. Since the authors look at the problem of node classification, it is very easy to find much more conventional examples with a more clear positive impact upon society (including for the graph rewiring scheme). Therefore, in my opinion, it would be best to change these examples entirely and find better ones to motivate the work, while still discussing negative applications in a societal impact section.
> >
> > However, if the authors still prefer to go with these examples, I believe they should describe it much more carefully and responsibly. For instance, as Reviewer 7ABz also mentioned, there is no need to mention the gender of the potential daters. The authors also mention in the introduction "tracking corruption-convictions among politicians" as another application, which seems a very peculiar example to me. which can also raise many issues regarding people's privacy and other related aspects. Consequently, I will also edit my initial review to specify that an ethics review is needed.

---

> > > ### Author Response · Authors · 2021-08-19
> > > **Response to Technical Questions (OGB Experiments, k+1 dimensional clique complexes, Computational complexity, and Ethical issues) (1/2)**
> > >
> > > Thank you very much for providing the additional questions! We have been waiting for the feedback and are particularly interested in the robustness and ethics questions that inspired some new directions to explore.
> > >
> > > Q1: “However, there it might be more appropriate to also include the relative drop in performance for each method with respect to the non-perturbed setting. Since your method starts from an initial score that is much higher than the baselines, the fact that your method ends up with a higher absolute score at different perturbation levels does not necessarily mean it is more robust. In fact, percentage-wise, on a first look, it seems that the performance of the proposed method degrades slightly more than some other baselines.”
> > >
> > > ${\bf A}$: Thank you for the interesting point which is tightly linked to robust statistics. Following your question, we have now provided both the decay rates compared with the initial accuracy, i.e., initial decay rate (IDR), and the relative decay rate (RDR). IDR is a difference between the accuracy on clean data and accuracy at the perturbation rate $p$. RDR is a difference between the accuracy on perturbed data at the perturbation rate $p$ and the accuracy on perturbed data at the perturbation rate $q$, where $p<q$. RDR is closely linked to the influence function/sensitivity curve in statistics as a standard measure of robustness, and IDR is somewhat related to gross-error shift sensitivity in robust statistics.
> > >
> > > Tables 3 and 4 below show IDR and RDR for random attacks on the OGB data and nettacks on Cora-ML. For the sake of room, the standard deviations are omitted (they are stated in the earlier responses, see the feedback of Q3 for Reviewer oJSD).
> > >
> > > We find that the decrease in IDR on OGB under random attacks is higher for the top performers TRI-GNN and APPNP at 0.05 perturbation rate. At higher perturbation rates, the less performing models (on clean graphs) lose more  in terms of IDR. IDR of TRI-GNN is lower than that of its next competitor APPNP for higher perturbations. In terms of RDR, which is an analogue of “derivative” and is related to the influence function in robust statistics, we find that TRI-GNN yields the lowest RDR for moderate perturbation rates (0.10) and competitive RDR for higher perturbation rate (0.25).  On nettacks on Cora-ML, TRI-GNN $\textbf{substantially}$ outperforms all other models in IDR and yields one of the best results in RDR, especially for moderate and higher perturbation rates. Also, Figure 3 in the main paper and Figure 3 in Appendix present the decay rates under various perturbations, clearly, the decay of TRI-GNN appears to be the less steeper.
> > >
> > > Hence, TRI-GNN has an advantage not only in terms of delivering high performance on clean graphs but also being robust.
> > >
> > > Important observation we make is that decay rates under attacks are nonlinear (as it might be expected).
> > >
> > > We will run a full study on IDR and RDR for other data and types of attacks. We suspect that TRI-GNN might be particularly good for targeted attacks.
> > >
> > > Table 3: Node classification performance on ogbn-products (Accuracy, RDR, and IDR) (%) under random attack.
> > > +++++++++++++++++++++++++++++++++++++++++++++++++++++++++++++
> > >
> > > Ratio of Noise Edges (%) | TRI-GNN$_\text{D}$ | GCN | GAT | RGCN | APPNP
> > >
> > > +++++++++++++++++++++++++++++++++++++++++++++++++++++++++++++
> > >
> > > 0 $\qquad$ $\qquad$ $\qquad$ $\qquad$|  $\quad$  84.00 $\quad$ | 78.87 | 80.02 | 81.35 | 83.17
> > >
> > > +++++++++++++++++++++++++++++++++++++++++++++++++++++++++++++
> > >
> > > 0.05 $\quad$ $\qquad$ $\qquad$ $\qquad$|  $\quad$ 78.77  $\quad$  | 75.74 | 77.73 | 78.35 |78.15
> > >
> > > RDR $\quad$ $\qquad$ $\qquad$ $\qquad$|  $\quad$ 5.23$\qquad$  |  3.13 | $\quad$  2.29| 3.00    | 5.02
> > >
> > > IDR $\quad$ $\qquad$ $\qquad$ $\qquad$ |  $\quad$ 5.23 $\qquad$| 3.13 | $\quad$  2.29| 3.00       | 5.02
> > >
> > > +++++++++++++++++++++++++++++++++++++++++++++++++++++++++++++
> > >
> > > 0.10 $\quad$ $\qquad$ $\qquad$ $\qquad$ | $\quad$  76.90 $\quad$  | 71.62 | 73.22 | 75.60 |76.10
> > >
> > > RDR $\quad$ $\qquad$ $\qquad$ $\qquad$ |$\quad$  $\quad$  1.87 $\quad$| 4.12 | $\quad$  4.51 | 2.75 | 2.05
> > >
> > > IDR $\quad$ $\quad$ $\quad$ $\qquad$ $\qquad$ | $\quad$  $\quad$  7.10$\quad$ | 7.25 | $\quad$  6.80 | 5.75 | 7.07
> > >
> > > +++++++++++++++++++++++++++++++++++++++++++++++++++++++++++++
> > >
> > > 0.25 $\quad$ $\qquad$ $\qquad$ $\qquad$ | $\quad$  71.50 $\quad$  | 68.17 | 70.00 | 69.59 |70.01
> > >
> > > RDR $\quad$ $\qquad$ $\qquad$ $\qquad$| $\quad$  5.40 $\qquad$| 3.45 | 3.22$\quad$ | 6.01 | 6.09
> > >
> > > IDR $\quad$ $\quad$  $\quad$ $\qquad$ $\qquad$| $\quad$  12.50 $\quad$  | 10.70 | 10.02 | 11.76 | 13.16
> > >
> > > +++++++++++++++++++++++++++++++++++++++++++++++++++++++++++++
> > >
> > >
> > > Table 4: Node classification performance (Accuracy, RDR, and IDR) (%) under nettack on Cora-ML.
> > > ++++++++++++++++++++++++++++++++++++++++++++++++++++++++++++++++
> > >
> > > \# of Perturbations Per Node | TRI-GNN$_{\text{A}}$ | GAT | RGCN | APPNP
> > >
> > > ++++++++++++++++++++++++++++++++++++++++++++++++++++++++++++++++
> > >
> > > 0 $\qquad$$\qquad$ $\qquad$ $\qquad$ $\qquad$| $\quad$  84.80 | 83.11 | 82.80 | 83.31
> > >
> > > ++++++++++++++++++++++++++++++++++++++++++++++++++++++++++++++++
> > >
> > > 1 $\qquad$$\qquad$ $\qquad$ $\qquad$ $\qquad$| $\quad$ 84.02 | 77. 02 | 77.29 | 78. 25
> > >
> > > RDR $\quad$ $\quad$ $\quad$ $\quad$ $\qquad$ $\quad$ $\quad$| $\quad$ 0.78 | $\quad$ 6.09 | 5.51 | 5.06
> > >
> > > IDR $\quad$ $\quad$ $\quad$ $\quad$ $\qquad$ $\quad$ $\quad$ | $\quad$ 0.78 | $\quad$ 6.09 | 5.51 | 5.06
> > >
> > > ++++++++++++++++++++++++++++++++++++++++++++++++++++++++++++++++
> > >
> > > 2 $\qquad$$\qquad$ $\qquad$ $\qquad$ $\qquad$| $\quad$79.77 | 74.00 | 73.52 | 75.23
> > >
> > > RDR $\quad$ $\quad$ $\quad$ $\quad$ $\qquad$ $\quad$ $\quad$| $\quad$ 4.25 $\quad$| 3.02 | 3.77 | 3.02
> > >
> > > IDR $\quad$ $\quad$ $\quad$ $\quad$ $\qquad$ $\quad$ $\quad$| $\quad$ 5.03$\quad$| 9.11 | 9.28 | 8.08
> > >
> > > ++++++++++++++++++++++++++++++++++++++++++++++++++++++++++++++++
> > >
> > > 3 $\qquad$$\qquad$ $\qquad$ $\qquad$ $\qquad$| 75.21 | 66.26 | 71.21 |68.00
> > >
> > > RDR $\quad$ $\quad$ $\quad$ $\quad$ $\qquad$ $\quad$ $\quad$| 4.56 $\quad$| 7.74 | 2.31 | 7.23
> > >
> > > IDR $\quad$ $\quad$ $\quad$ $\quad$ $\qquad$ $\quad$ $\quad$|9.59 $\quad$| 16.85 | 11.59 | 15.31
> > >
> > > ++++++++++++++++++++++++++++++++++++++++++++++++++++++++++++++++
> > >
> > > 4 $\qquad$$\qquad$ $\qquad$ $\qquad$ $\qquad$| 70.25 |63.56 | 63.89 |65.07
> > >
> > > RDR $\quad$ $\quad$ $\quad$ $\quad$ $\qquad$ $\quad$ $\quad$| 4.96 $\quad$| 2.70 | 7.32 | 2.93
> > >
> > > IDR $\quad$ $\quad$ $\quad$ $\quad$ $\qquad$ $\quad$ $\quad$|14.55$\quad$ | 19.55 | 18.91 | 18.24
> > >
> > > ++++++++++++++++++++++++++++++++++++++++++++++++++++++++++++++++
> > >
> > > 5 $\qquad$$\qquad$ $\qquad$ $\qquad$ $\qquad$| 68.39 |61.79 | 58.50 | 62.11
> > >
> > > RDR $\quad$ $\quad$ $\quad$ $\quad$ $\qquad$ $\quad$ $\quad$| 1.86$\quad$ | 1.77 | 5.39$\quad$ | 2.96
> > >
> > > IDR $\quad$ $\quad$ $\quad$ $\quad$ $\qquad$ $\quad$  $\quad$|16.41$\quad$ | 21.32 | 24.30 | 21.20
> > >
> > > ++++++++++++++++++++++++++++++++++++++++++++++++++++++++++++++++

---

> > > > ### Comment · Reviewer_uhB4 · 2021-08-30
> > > > **New Response**
> > > >
> > > > __Considering the 2-clique complex for $k=1$__
> > > >
> > > > Thank you for the detailed response! I am not disagreeing with the authors here. I was merely pointing out the fact that a 2-clique complex is not absolutely necessary. Of course, not considering 2-simplices comes at a cost, as the authors have correctly pointed out. In any case, this point is not so important since the authors do not consider $k > 0$ in practice anyway. Therefore, I would just reiterate here my suggestion to remove all the stuff about simplicial complexes and other terminology required by $k > 1$ and perhaps parts of this response can be included in the appendix.
> > > >
> > > > __Computational Complexity__
> > > >
> > > > The authors still did not report the overall computational complexity in their answer for the whole model. Suppose the graph is fully connected, then the complexity becomes O(n^2) (looking at all the pairs of nodes) times O(n^3) (computing the PD + Wasserstein distance between the PDs produced by each pair of nodes). This results in an overall computational complexity of O(n^5), which is huge.
> > > >
> > > > Even under a milder assumption, suppose there exists a node that is connected to all the nodes in the graph. This automatically makes the complexity O(n^3) because of the Wasserstein distance computation, even if the graph is very sparse.
> > > >
> > > > __Robustness statistics__
> > > >
> > > > Thank you for adding these! I think they will significantly improve that section of the paper.
> > > >
> > > > __Proof__
> > > >
> > > > I had a minimal look over the new proof and it seems all right (although it is possible I might have missed something). Please include the technical version in the supplementary. I think this is absolutely necessary. The intuitive explanations should complement the technical proof rather than replace it.
> > > >
> > > > At the same time, I also share the concerns of Reviewer srjf about the optimality of these bounds and how useful they are in practice. If K becomes significantly larger than the maximum average degree or equivalently, the number of nodes in $T^{\pm}$, the bound is not helpful at all.

---

> > > > > ### Author Response · Authors · 2021-08-31
> > > > > **Response to New Questions**
> > > > >
> > > > > Thank you very much for the additional feedback! We have been waiting for it and appreciate your response.
> > > > >
> > > > > Q1: “Considering the 2-clique complex for $k=1$.”
> > > > >
> > > > > ${\bf A}$: On ${\bf 2}$-$\bf clique$ $\bf complexes$ $\bf and$ $\bf{robustness}$, yes, we will certainly add these details and revisions into the final version.
> > > > >
> > > > > Q2: “Computational Complexity.”
> > > > >
> > > > > ${\bf A}$: On ${\bf computational}$ $\bf{complexity}$, yes, for the case of the complete (i.e., fully connected graph), the overall complexity will be $O(n^5)$, where $n$ is the number of nodes. However, a complete graph is a far extreme case.
> > > > >
> > > > > In less extreme scenarios, the expected number of nodes in the $k$-hop neighborhood can be approximated by $\bar{d}^k$, where $\bar{d}$ is the mean degree of the graph and mathematical expectation is in terms of the node degree distribution. The computational complexity for Wasserstein distance between two persistence diagrams is $O(m^3)$, where $m$ is the total number of barcodes in the both persistence diagrams [Chen and Wang (2021) “Approximation algorithms for 1-Wasserstein distance between persistence diagrams”]. In the worst case scenario, if we use a very fine filtering function for persistence diagrams of $k$-hop neighborhoods, it would give at most $2\bar{d}^k$ barcodes (i.e., the number of barcodes in each $k$-hop neighborhood is at most $\bar{d}^k$ and we compare two $k$-hop neighborhoods). Considering we use the degree filtering function for our persistence diagrams, we expect to have much less ($\sim \bar{d}^{\frac{k}{2}}$) barcodes in the $0^{th}$ persistence diagram of a $k$-hop neighborhood by using the technique in [Kanari et al. (2020) “From trees to barcodes and back again: theoretical and statistical perspectives”]. Hence, the overall computational complexity would be of stochastic order $O(n^2 \bar{d}^{\frac{3k}{2}})$. Yes, it may be costly to run persistent homology (PH) for denser graphs (currently, computational complexity is the main roadblock for all PH-based methods on graphs) but as we noted earlier, for denser graphs we  may use a degree filtration under lower $k$-hops which is still found to be very powerful even for small $k$.
> > > > >
> > > > > Q3: “Proof.”
> > > > >
> > > > > ${\bf A}$: On the $\bf{proof}$, we do not claim that this is an optimal bound. However, it still provides an important theoretical relationship between average degrees and persistent homologies of the original and distorted graphs and has multiple implications for future joint analysis of graph algebraic connectivity and graph PH.  We understand your concern that taking a sufficiently large constant $K$ could make the theorem pointless. However, even though our constant $K$ is not optimal, the one we obtain in the proof meaningfully relates the distance between the induced persistent diagrams to the average degree of the induced rewired graphs as follows: By using the same notations as in the technical proof above, you can see that our constant $K(t)$ comes from the quantity $\frac{k^+(t)-k^+(t-2\delta)}{\delta}$, where $k^\pm(t)$ is the number of edges to be added to graph $G^\pm$ when $\delta$ is the PD distance between $G^+$ and $G^-$.  Since we explicitly state a lower bound for nontrivial $\delta$ as the minimum distance between the threshold pairs, the quantity $K(t)$ given in the proof cannot be very large, and is basically controlled by the quantity $k^+(t)-k^+(t-2\delta)$. Hence, for small $t=\epsilon_1$, $K(\epsilon_1)$ will be a very small number which proves our main point in the theorem: When the graphs are topologically similar (when $\delta=\mathcal{D}(G^+,G^-)$ small), then the average degrees of the induced rewired graphs $T^+$ and $T^-$ are very close to each other.
> > > > > With the currently available machinery of persistent homology on graphs, obtaining a sharp bound for such inequality is infeasible, given that such bound depends on a particular filtering function and there are yet no theoretical results on robustness of Wasserstein distance among two PDs on the graph to minor perturbations. However, our inequality paves a path for derivation of some new theoretical results on robustness of Fiedler eigenvalues and statistical inference for topological graph invariants.

---

> > > > > > ### Comment · Reviewer_uhB4 · 2021-09-02
> > > > > > **Last Response**
> > > > > >
> > > > > > Thank you for these clarifications! I have eventually decided to raise my score by one point (i.e. 7).
> > > > > >
> > > > > > Re bound optimality: Could you add some numerical experiment plotting as a function of epsilon the theoretical upper bound versus the observed quantity for multiple graph datasets? Adding the maximum possible value of the difference between the average degrees as a horizontal bar to these plots could also be useful as a dummy baseline for the provided upper bound and would show in which epsilon regimes it is useful. Since the theoretical analysis does not answer these questions, the numerical experiments could complement it. I think this plot could strengthen the case for the Theorem.
> > > > > >
> > > > > > However, I understand this request comes quite late now in the review process so I do not expect the authors to implement it now, but it would be something useful for the final version of the paper.

---

> > > > > > > ### Author Response · Authors · 2021-09-02
> > > > > > > **Feedback**
> > > > > > >
> > > > > > > Thank you very much for your positive feedback and also for the very motivating discussion throughout the review period!
> > > > > > >
> > > > > > > Yes, we will add such a plot to the final version (we cannot add any plots to our responses on openreview unless we create an external link and it will also take some time to prepare the plot).

---

> > > ### Author Response · Authors · 2021-08-19
> > > **Response to Technical Questions (OGB Experiments, k+1 dimensional clique complexes, Computational complexity, and Ethical issues) (2/2)**
> > >
> > > Q2: “Persistence homology is still well-defined for k=1 without considering the 2-clique complex. In that case, it should capture information about all the cycles in the graph.”
> > >
> > > ${\bf A}$: Yes, persistence homology is still well-defined, but if one does not use $(k+1)$ dimensional simplices in the construction, formally from an algebraic topology point of view, it can hardly be called homology anymore. In algebraic topology, $k$-homology is to count $k$-dimensional "essential" topological features. That is, it counts $k$-dimensional closed (possibly singular) submanifolds, which $\textbf{do not bound}$ any $(k+1)$-domains. If we do not use any $(k+1)$ dimensional domain in the construction, it basically counts all $k$-cycles whether filled or non-filled. Such case sometimes is referred to as non-regular homology, while the case we discuss is referred  to as regular homology.
> > >
> > > In dimension $k=1$, you are right that some of the authors set the maximum dimension to $1$ in the filtration construction. That means they not only count the "genuine" cycles (real holes) in the induced simplicial complex, but all triangles in the graph, too. The main reason for that is when the graph is sparse, many filtration steps do not produce enough $1$-dimensional genuine cycles to differentiate between other graphs. When you set maximum dimension to $1$, you start counting all triangles, and other trivial cycles, too. In a sense, this count gives you some sort of density measurement. However, it is no longer relevant to homology in the standard pure math sense.
> > >
> > > On the other hand, in this case, the computation of persistence diagrams becomes trivial by a simple Euler Characteristics argument. For a graph $\mathcal{G}=(\mathcal{V},\mathcal{E})$ , $\chi(\mathcal{\mathcal{G}})=\beta_0(\mathcal{G})-\beta_1(\mathcal{G})=\mathcal{V}-\mathcal{E}$ where $\beta_i$ is the $i^{th}$ Betti number (rank of the homology group $H_i(\mathcal{G})$). Here, if we do not include $2$-dimensional simplices in the simplicial complex, there would not be any $\beta_i$ for $i\geq 2$, and the Euler Characteristics equation follows. This implies, without $2$-simplices, first homology of the graph can simply be expressed in terms of number of components ($\beta_0(\mathcal{G})$), the number of vertices $\mathcal{V}$ and the number of edges $\mathcal{E}$, i.e., $\beta_1(\mathcal{G})=\beta_0(\mathcal{G})-\mathcal{V}+\mathcal{E}$. This is the count of trivial and nontrivial cycles in the graph which can be simply obtained by edge counts as above. Since there are no $2$-simplices in the complex, no $1$-cycle can die in the filtration. Hence, the death times for all barcodes will be $\infty$. The birth times of all barcodes can be found via $\beta_1(\mathcal{G}_k)$ where $\{\mathcal{G}_k\}$ are the subgraphs in the filtration sequence.
> > >
> > > In technical terms, one can see the need for $(k+1)$-dimensional simplices to obtain "genuine" $k$-dimensional topological features as follows. Let $\mathbf{Y}$ be a simplicial complex. Let $\mathcal{C}_k(\mathbf{Y})$ represent the $k$-dimensional chains of $\mathbf{Y}$. Then, we have the following induced sequence
> > >
> > > \begin{equation}
> > > \dots\xrightarrow{\partial_{k+2}} \mathcal{C}_{k+1}\(\mathbf{Y}\)\xrightarrow{\partial_\{k+1\}}\mathcal{C}_k(\mathbf{Y})\xrightarrow{\partial{k}}\mathcal{C}_\{k-1\}(\mathbf{Y})\xrightarrow{\partial_\{k-1\}}\dots
> > > \end{equation}
> > >
> > > where $\partial_k$ is the boundary map. Then, the homology is defined by taking a quotient of all $k$-cycles via all $k$-boundaries. In particular, we call $Z_k(\mathbf{Y})=ker(\partial_k)\subset \mathcal{C}_{k}(\mathbf{Y})$ $k$-cycles of $\mathbf{Y}$.
> > >
> > > Similarly, we define $B_k(\mathbf{Y})=im(\partial_{k+1})\subset \mathcal{C}_{k}(\mathbf{Y})$  $k$-boundaries of $\mathbf{Y}$. Here, $k$-cycles $Z_k(\mathbf{Y})$ represents all $k$-dimensional (possibly singular) closed submanifolds, while $k$-boundaries  $B_k(\mathbf{Y})$ represents all $k$-cycles which are filled by a $(k+1)$-domain in $\mathbf{Y}$. In other words, $B_k(\mathbf{Y})$ captures the information of "trivial" $k$-cycles. Then, the $k$-homology is defined by taking quotient of all $k$-cycles $Z_k(\mathbf{Y})$ by the trivial ones $B_k(\mathbf{Y})$, i.e. $H_k(\mathbf{Y})=Z_k(\mathbf{Y}) / B_k(\mathbf{Y})$. Therefore, if we do not use $(k+1)$-dimensional simplices in the construction, then $\mathcal{C}_\{k+1\}(\mathbf{Y})$ would be trivial, and so is $B_k(\mathbf{Y})=im(\partial_\{k+1\})$. This implies the "homology" computation in such setting would automatically $H_k(\mathbf{Y})=Z_k(\mathbf{Y})$. However, in such setting without $(k+1)$-simplices, this no longer represent "homology" invariant in the usual sense, and its computation is very simple with a basic Euler Characteristics argument as shown above. For further details [Munkres, James R. Elements of algebraic topology. CRC press, 2018.]
> > >
> > > Q3: “The authors have only included the running time in the paper, not the computational complexity. The complexity of  $O(mlog(m))$  is only for computing persistence homology in dimension zero. The complexity increases for $k>0$  and I believe the authors use such values. Furthermore, the method performs lots of additional computations besides computing the persistent homology. For instance, it computes the Wasserstein distance between persistence diagrams. What is the complexity associated with that? What is the complexity associated with the computation of $W^{topo}$? Finally, I would like to clearly see the computational complexity of all these individual components and the resulting overall complexity. This analysis should also take into account various hyper-parameters that might affect the complexity (like the dimension of the PD).”
> > >
> > > ${\bf A}$: For d-dimensional persistent homology calculations, the computational complexity is $O(m^{d})$ (that is why we mainly have considered 0-dimensional topological feature in our experiment and we also have found that performance gains due to including 1-dimensional features are negligible (see the feedback for Q1(Reviewer srjf))). The computational complexity of computing Wasserstein distance between two persistence diagrams (PDs) with $n$ points is $O(n^3)$. To obtain the final $W^{topo}$, we sum up the original adjacency matrix $W$, positive topology-induced adjacency matrix $W^{topo^{+}}$, and negative topology-induced adjacency matrix $W^{topo^{-}}$, where positive topology-induced and negative topology-induced adjacency matrices are generated via setting thresholds for existing Wasserstein distance information (see Eq.2 in the paper).
> > >
> > > Q4: “First of all, apologies for not mentioning this in my first response! I have made a note of it while reading the paper but somehow I forgot to transcribe it in my response here. However, I noticed that other reviewers have rightly flagged some possible ethical issues around the examples used in the paper.
> > > In my opinion, the examples chosen by the authors are not entirely appropriate and touch upon some very sensitive subjects. Since the authors look at the problem of node classification, it is very easy to find much more conventional examples with a more clear positive impact upon society (including for the graph rewiring scheme). Therefore, in my opinion, it would be best to change these examples entirely and find better ones to motivate the work, while still discussing negative applications in a societal impact section. However, if the authors still prefer to go with these examples, I believe they should describe it much more carefully and responsibly. For instance, as Reviewer 7ABz also mentioned, there is no need to mention the gender of the potential daters. The authors also mention in the introduction "tracking corruption-convictions among politicians" as another application, which seems a very peculiar example to me which can also raise many issues regarding people's privacy and other related aspects. Consequently, I will also edit my initial review to specify that an ethics review is needed.”
> > >
> > > ${\bf A}$: We particularly appreciate these questions on ethics as they pushed us into a completely uncharted territory of TDA utility and limitations for fairness in ML.
> > > We agree that the stated examples may touch quite sensitive subjects though we believe that virtually any problem involving real data may lead to both positive and negative outcomes. The reason why we have selected such (unconventional) example is that we tried to show a scenario (more personal and closer to many people) where the peer-effect of the neighborhood matters. Also, ML is still largely under-used in dating services, and topological tools have never been used before for such application.
> > >
> > > One other example is the customer churn and retention analytics based on peer effects. That is, the goal here is to classify the node as potential churn customer based not only on the individual attributes but on interactions among its neighbor attributes. Positive societal impacts include pro-active strategies the company can undertake to retain the client, higher profit, and better, more targeted customer service. Negative societal impacts include the potential bias toward specific groups of customers from certain under-represented subpopulations (here node classification based on local topological features may demonstrate even higher bias than other non-neighborhood based methods, as data bias might be amplified due to bias in peers data). This is indeed a highly negative aspect that needs to be explored and mitigated.
> > >
> > > We will revise the paper and discuss positive and negative aspects of the examples and their implications for usage of TRI-GNN  in the final version.

---

> ### Author Response · Authors · 2021-08-06
> **Response to Technical Questions (1/2: Q1, Q2, Q3, Q4, Q5, Q6)**
>
> Q1: "I believe the main limitation of the work is that the experimental evaluation focuses only on a set of datasets with very small graphs. While these datasets could also provide valuable information about a method, it only shows how a method performs in a very limited setting. The limitations of these datasets have been widely studied recently (e.g. https://arxiv.org/abs/2003.00982). Therefore, I wished the authors complemented these experiments with (some) benchmarks from OGB (https://ogb.stanford.edu/) or the Benchmarking GNNs paper (https://arxiv.org/abs/2003.00982). These limitations also transfer over to the perturbation stability experiments."
>
> ${\bf A}$: Thank you for suggesting these papers and datasets! We are currently running experiments and will post our results soon.
>
> Q2: "The introduction to persistent homology is probably too abrupt for the average reader, who is likely not familiarised with persistent homology. For instance, the authors mention a simplicial complex when introducing PH, but simplicial complexes are never defined. Furthermore, I believe they are not even needed since this work is only interested in graphs (i.e. 1-dimensional simplicial complexes). At the same time, the authors mention k-dimensional topological features. But again, since we are talking about graphs we are not interested in k > 1. So these terms can be completely excluded to avoid confusing people and maintain only a minimally required technical jargon. A recent brief introduction to PH that I liked can be found in the background section of https://arxiv.org/abs/2102.07835. Notice how the authors use various examples to build an intuition. Perhaps the authors could borrow some ideas from there in the way the background material can be presented."
>
> ${\bf A}$: Thank you very much for bringing this up! In general, k-dimensional topological features can be useful especially in the case of sufficiently dense graphs for sublevel filtrations. Even though the general usage of persistent homology is mostly restricted to dimensions k=0 and k=1 because of the computational costs, it is known that the higher dimensional topological features can also be useful in general especially in the power filtration case (See Adamaszek Israel Journal of Mathematics2013, and  Adamaszek and Adams Pacific Journal of Mathematics2017).
>
> You are right that for k=0, one does not need clique complexes, but for k=1 and higher, one needs to take at least k+1 dimensional clique complexes (or different simplicial complexes) to define persistent homology (Aktas et al. Applied Network Science2019). Since this was a general background on persistent homology, we tried to keep it broad.
>
> Thank you very much for your suggestion to simplify the background, and for the reference. In the final version, we will restrict ourselves to the suggested more focused background which is just necessary to follow our construction.
>
> Q3: "On the latter point, many definitions lack an intuitive explanation or justification. For instance, the intuition behind the Wasserstein distance is never explained. Or the intuition/motivation behind the expression for Wuvtopo below line 264, which otherwise might seem a bit arbitrary."
>
> ${\bf A}$: We used Wasserstein distance to measure the topological similarities of two $k$-hop neighborhoods of the nodes. For each $k$-hop neighborhood, we obtained the persistence diagrams. Wasserstein distance is a distance defined on the space of persistence diagrams. By theory, if two spaces are topologically close  to each other (Gromov-Hausdorff distance), then their persistence diagrams are close to each other with respect to Wasserstein distance. Hence, if the Wasserstein distance between the persistence diagrams of the k-hop neighborhoods of the nodes $u$ and $v$ is small, we derive that these neighborhoods are topologically similar, and so the nodes are similar. In turn, $\epsilon_1$ and $\epsilon_2$ are the thresholds for this topological similarity measure to obtain rewired graphs.
>
> For $W_{uv}^{topo}$, you are right. We need to give a more intuitive explanation. Here $W_{uv}^{topo}$ is the adjacency matrix of the induced TIMR graph after rewiring. There are the matrices of same size, $W$, $W^{+}$, and $W^{-}$. $W$ is the original adjacency matrix. $W^{+}$ is the matrix for the edges ($a_{ij}=1$) to be added for $v_i$ and $v_j$ where the topological similarity $c$ (Wasserstein distance between persistence diagrams of the k-hop neighborhoods of $v_i$ and $v_j leq \epsilon_1$). So, $W^{+}$ keeps the information of the edges to be added between the topologically similar nodes. $W^{-}$ is the opposite one where topological similarity $\geq \epsilon_2$, then remove the edge. Meaning the entry $a_{ij}=-1$ for $v_i$ and $v_j$ if the topological similarity $\geq \epsilon_2$. Then, we add $W_{ij}$+ $W_{ij}^{+}$ + $W^{-}_{ij}$. If the result is $\geq 1$, then we keep the edge in the final graph. To sum up, if two nodes are topologically similar, then add an edge between them if there is none. If two nodes are topologically dissimilar, then remove the edge if there is one.
>
> Thank you very much for your suggestions! We will add more intuitive explanations to these definitions, and statements during revision.
>
>
> Q4: "While the Figures offer a pretty good visual description the captions are too brief and offer little information. For instance, the size of the arrows in Figure 2 and the colors of different elements are not explained at all."
>
> ${\bf A}$: We have provided detailed overview of Figures 1 and 2 in the supplementary material (see Section C; # line 21 - 26). We will update Figure captions in the final version.
>
> Q5: "The paper is overflown with acronyms (TIMR, STAN, TRI-GNN) and the authors do not explain very well how all these pieces are coming together. I've understood this only after noticing in Figure 1. However, the main text never provides a unifying view and the sections seem a bit disconnected because of this. The order they are presented in could also be improved. I was very confused by how STAN and TRI-GNN interact since both of them produce multiple layers of features. This is roughly clear from the Figure, but still, I feel like I am missing many details."
>
> ${\bf A}$: It’s a miss on our side. In the final version, we will add an overarching view that unifies all sections and discusses their interactions before the current section 4.1.
>
> Q6: "I do not think it was explained (apologies if I missed it), but how are the results for the baselines produced? Are they taken from other papers or have the authors trained them? If it is the latter, how where these baselines fine-tuned to ensure a fair comparison."
>
> ${\bf A}$: For citation networks, we reuse the default configuration given by the source code and paper; for power grid networks, we tune the hyperparameters via grid search and report the best results from the optimal hyperparameter configuration. We will report the set of all hyperparameters that we use to train the baselines in the final version.

---

> ### Author Response · Authors · 2021-08-10
> **Response to Technical Questions (Q1 - additional feedback)**
>
> Q1: "I believe the main limitation of the work is that the experimental evaluation focuses only on a set of datasets with very small graphs. While these datasets could also provide valuable information about a method, it only shows how a method performs in a very limited setting. The limitations of these datasets have been widely studied recently (e.g. https://arxiv.org/abs/2003.00982). Therefore, I wished the authors complemented these experiments with (some) benchmarks from OGB (https://ogb.stanford.edu/) or the Benchmarking GNNs paper (https://arxiv.org/abs/2003.00982). These limitations also transfer over to the perturbation stability experiments."
>
> ${\bf A}$: As suggested, we have conducted experiments on node classification (Table 1 below) and robustness (Table 2 below) on two large OGBN datasets, i.e., ogbn-products (# nodes = 2,449,029, # edges = 61,589,140) and ogbn-arxiv (# nodes = 169,343, # edges = 1,166,243). The data split rates for ogbn-products and ogbn-arxiv are 10%/2%/88% and 78%/8%/14% respectively. Table 1 below shows that our proposed TRI-GNN$_\text{D}$ always outperforms state-of- the-art methods in terms of classification accuracy on large networks. The results also indicate that our proposed model is capable to achieve highly promising results on very large networks.
>
> Table 1: Node classification results on ogbn-products and ogbn-arxiv.
>
> +++++++++++++++++++++++++++++++++++++++++++++++++++++++++++++++++++++++++++
>
> Dataset $\quad$ $\quad$ | TRI-GNN$_\text{D}$ $\quad$ | $\quad$ GCN $\quad$ | $\quad$ $\quad$ GAT $\quad$ | $\quad$ RGCN $\quad$ | APPNP
>
> +++++++++++++++++++++++++++++++++++++++++++++++++++++++++++++++++++++++++++
>
> ogbn-products | 84.00 (0.007) | 78.87 (0.003) | 80.02 (0.001) | 81.35 (0.005) | 83.17 (0.006)
>
> +++++++++++++++++++++++++++++++++++++++++++++++++++++++++++++++++++++++++++
>
> ogbn-arxiv $\quad$  | 74.30 (0.003) | 72.18 (0.002) | 72.47 (0.002) | 72.97 (0.001) | 72.10 (0.003)
>
> +++++++++++++++++++++++++++++++++++++++++++++++++++++++++++++++++++++++++++
>
> We also have performed the perturbation (i.e., random attacks) stability experiments on the ogbn-products. From Table 2, we observe that our proposed TRI-GNN$_\text{D}$ consistently outperforms other baselines under different ratios of  noise edges. This result validates robustness of our proposed TRI-GNN$_\text{D}$ even over very large networks.
>
> Table 2: Node classification performance on ogbn-products (Accuracy $\pm$ Std) under random attack.
>
> ++++++++++++++++++++++++++++++++++++++++++++++++++++++++++++++++++++++++++++++++++++
>
> Ratio of Noise Edges (%) | TRI-GNN$_\text{D}$ $\quad$ | $\quad$  GCN $\quad$  | $\quad$  $\quad$  GAT $\quad$  | $\quad$  RGCN $\quad$   | $\quad$ APPNP
>
> ++++++++++++++++++++++++++++++++++++++++++++++++++++++++++++++++++++++++++++++++++++
>
> 0 $\qquad$ $\qquad$ $\qquad$ $\qquad$| 84.00 (0.007) | 78.87 (0.003) | 80.02 (0.001) | 81.35 (0.005) | 83.17 (0.006)
>
> ++++++++++++++++++++++++++++++++++++++++++++++++++++++++++++++++++++++++++++++++++++
>
> 0.05 $\quad$  $\qquad$ $\qquad$ $\qquad$| 78.77 (0.013) | 75.74 (0.008) | 77.73 (0.003) | 78.35 (0.015) | 78.15 (0.022)
>
> ++++++++++++++++++++++++++++++++++++++++++++++++++++++++++++++++++++++++++++++++++++
>
> 0.10 $\quad$ $\qquad$ $\qquad$ $\qquad$| 76.90 (0.020) | 71.62 (0.011) | 73.22 (0.009) | 75.60 (0.020) | 76.10 (0.039)
>
> ++++++++++++++++++++++++++++++++++++++++++++++++++++++++++++++++++++++++++++++++++++
>
> 0.25 $\quad$ $\qquad$ $\qquad$ $\qquad$| 71.50 (0.015) | 68.17 (0.009) | 70.00 (0.007) | 69.59 (0.018) | 70.01 (0.028)
>
> ++++++++++++++++++++++++++++++++++++++++++++++++++++++++++++++++++++++++++++++++++++

---

### Official Review · Reviewer_BcEA · 2021-07-15

**Rating:** 5
**Confidence:** 4

**Summary:**

To tackle the over-smoothing problem in GNNs, this work proposes to use topological relational inference for systematically learning local graph structures. The similarity between two local graphs are measured by the persistent homology. A “multiedge” is added between two nodes whose local graphs are similar enough.  The local topological side information is added enrich the features of each node. Theoretical stability guarantees were derived for the above new graph representation. The experimental results provide empirical evidence to support the proposed approach.

**Ethical Concerns:**

No.

**Limitations And Societal Impact:**

No potential negative societal impact

**Main Review:**

Originality: The idea of using persistent homology to define neighborhood similarity between nodes is relatively novel. It may be a new direction for better understanding the graph algebraic connectivity.

Significance: A nice attempt to better utilize local topological information. Derive theoretical stability guarantees for the new local topological representation of graph.

Quality: I have several concerns. The way of constructing the ‘joint topology-induced adjacency matrix’ needs further consideration. The topology-induced adjacency matrix somehow tangles the connection information and the topological side information (eq. (3) in Section 4.4). STAN uses traditional neighborhood aggregations except for a different measurement in node distance, which make it limited. Although the proposed approach outperforms some baseline models on the Cora, CiteSeer, and PubMed datasets, it remains limited compared to other GNN models (e.g., NodeNet and GCN-label propagation). For example, NodeNet achieves around 86% accuracy on node classification tasks on Cora.

Clarity: The first several sections of this paper are OK. The model, implementations, and experiment sections need improvements. The explanations about TDA can be improved. As it is written now, it is hard for readers to understand the idea without enough background knowledge. There are several places that need some clarifications. For example, how sequence {Gi} induces the filtration of complex {Ci}, how to decide the value of k, the similarity between graph {Gi} and {Gi+1}, and so on. How to deal with continuous attributes in the attribute-based filtration of persistent homology?


**Time Spent Reviewing:**

23

---

> ### Author Response · Authors · 2021-08-06
> **Response to Technical Questions (1/2)**
>
> Q1: "I have several concerns. The way of constructing the ‘joint topology-induced adjacency matrix’ needs further consideration. The topology-induced adjacency matrix somehow tangles the connection information and the topological side information (eq. (3) in Section 4.4). STAN uses traditional neighborhood aggregations except for a different measurement in node distance, which make it limited. Although the proposed approach outperforms some baseline models on the Cora, CiteSeer, and PubMed datasets, it remains limited compared to other GNN models (e.g., NodeNet and GCN-label propagation). For example, NodeNet achieves around 86% accuracy on node classification tasks on Cora."
>
> ${\bf A}$: Thank you for suggesting NodeNet and GCN-LPA! We are currently running experiments on these models will post our results soon.
>
> The idea of the topology-induced adjacency matrix (eq. (3) in Section 4.4) is to reward or penalize connectivity among two nodes based on how similar shapes (i.e., topological side information) of their neighborhoods are.
>
> While we use neighborhood aggregation based on topological distances in STAN, the key machinery of the local node neighborhood PH can be extended, e.g., to topological attention mechanism with the LeakyReLU nonlinearity, PH-induced graph diffusion and other related tools.
>
> Q2: The first several sections of this paper are OK. The model, implementations, and experiment sections need improvements. The explanations about TDA can be improved. As it is written now, it is hard for readers to understand the idea without enough background knowledge. There are several places that need some clarifications. For example, how sequence {Gi} induces the filtration of complex {Ci}, how to decide the value of k, the similarity between graph {Gi} and {Gi+1}, and so on. How to deal with continuous attributes in the attribute-based filtration of persistent homology?
>
> ${\bf A}$: Thanks for bringing this up! We will give more details on the background in the revision. After constructing the “increasing” sequence of subgraphs $\{G_i\}$, there are several ways to obtain simplicial complexes. The most common one is to take the clique complex of $G_i$. Clique complex $C_i$ is a simplicial complex obtained by filling each complete $k$-subgraph ($k$-clique) with a ($k-1$)-dimensional simplex. This process naturally induces a sequence of simplicial complexes $\{C_i\}$, and persistent homology keeps track of the topological features which are born, surviving and dying in this sequence.
> The length $k$ of the sequence depends on the graph, and how you construct the sequence. For example, if one takes sublevel filtration for the degree function on the graph, then $k$ would naturally be the maximum degree of the nodes in the graph. If one uses a different function, or edge weights, then the length of the sequence $k$ (the number of the thresholds for the function values or edge weights) depends on how fine topological information you want to get. If this function is continuous, usually one chooses equally distributed thresholds.
>
> The similarity of the subgraphs in the sequence $G_i$ and $G_{i+1}$ is completely determined by how you construct this sequence. If you use the degree function, then $G_i$ only contains the nodes with degree $\leq i$, and edges between these nodes.
>
> Per your suggestion, we will give more details in the background of the setup of persistent homology.

---

> > ### Comment · Reviewer_BcEA · 2021-08-10
> > **Clarifications needed**
> >
> > The symbol "k" is used in multiple context, for example, k-subgraph, k-dimension, k-hop, k-subgraph (clique), and so on. It is confusing. It is not clear "Clique complex C_i is a simplicial complex obtained by filling each complete k-subgraph (k-clique) with a (k-1)-dimensional simplex".

---

> > > ### Author Response · Authors · 2021-08-11
> > > **Clarifications of the Symbol $k$**
> > >
> > > Q: "The symbol "k" is used in multiple context, for example, k-subgraph, k-dimension, k-hop, k-subgraph (clique), and so on. It is confusing. It is not clear "Clique complex C_i is a simplicial complex obtained by filling each complete k-subgraph (k-clique) with a (k-1)-dimensional simplex"."
> > >
> > > ${\bf A}$: We are sorry for the confusion. Indeed, there is some abuse of notations which we try to clarify below along with the key ideas behind persistent homology.
> > >
> > > 1 -  $\bf{k}$ ${\textbf{as in the clique construction}}$: Let us change it to $m$ for now to avoid confusion. A complete $m$-subgraph is the graph with $m$-nodes where every node is connected to every other node with an edge. Another term for complete $m$-subgraph is $m$-clique. In turn, the clique complex is an abstract simplicial complex (i.e, a family of finite sets closed under the operation of taking subsets), formed by the sets of nodes in the cliques. To obtain a clique complex of the graph, we represent each $m$-clique  as an $(m-1)$-dimensional simplex. For example, 1-cliques correspond to nodes (i.e., 0-dimensional simplices); 2-cliques correspond to edges (i.e., 1-dimensional simplices); 3-cliques correspond to triangles (with filled in interior) (i.e., 2-dimensional simplices);  $4$-clique corresponds to a tetrahedron (i.e.,  $3$-dimensional simplices).
> > >
> > > Now, note that  any subset of a clique is itself a clique, hence, it satisfies the definition of an abstract simplicial complex. As result, by placing each $m$-clique in correspondence with an $(m-1)$-dimensional simplex, we obtain a simplicial complex called the clique complex of the graph.
> > >
> > > 2- $\bf k$ ${\textbf{as the length of the filtration sequence}}$ (line 117): Let us change $k$ to $\mathcal{N}$ for now. Then, the length $\mathcal{N}$ of the filtration is exactly the number of thresholds $(\alpha_1<\alpha_2<\dots<\alpha_\mathcal{N}$) we use in the construction of our sequence $G_1\subset G_2 \subset \dots \subset G_\mathcal{N}$. This choice depends on the selected filtration function and on how fine topological information we aim to obtain. For example, if we use a discrete function (like node degree function), then the range of the function determines $\mathcal{N}$. The maximum $\mathcal{N}$ will be the maximum node degree in this example. If we use a continuous filtration function  (e.g., edge weights in $R^+$) to construct the sequence, then the number of weight cuts are determined by the user depending on how fine information he/she wants get. The higher the length $\mathcal{N}$, the finer topological information we obtain. However, for computational limitations, $\mathcal{N}$ is mostly chosen between 10-20 range, and typically equally distributed thresholds are used.
> > >
> > > 3 - $\bf k$ ${\textbf{as in the $k$-hop neighborhood}}$ (line 158) ${\textbf{and its selection}}$.  When we say $k$-hop neighborhood, we mean $k$-neighborhood of a node in the graph, i.e the subgraph induced by the nodes whose distance to the original node $\leq k$. The choice of $k$ is balanced by two goals: to collect sufficiently rich topological information from each node neighborhood and to maintain computational costs at a lower level. In our analysis we use cross validation, and as expected, if the average degree of the ambient graph is low, then we tend to choose larger $k$, while if the average degree is high, we choose $k$ small.
> > >
> > >
> > > Thank you for bringing up this confusion to our attention. We will clarify them in the revision.

---

> ### Author Response · Authors · 2021-08-10
> **Response to Technical Questions (2/2)**
>
> Q1: "Although the proposed approach outperforms some baseline models on the Cora, CiteSeer, and PubMed datasets, it remains limited compared to other GNN models (e.g., NodeNet and GCN-label propagation). For example, NodeNet achieves around 86% accuracy on node classification tasks on Cora."
>
> ${\bf A}$: ${\textbf{Summary}}$: NodeNet (Dabhi and Parmar arxiv2020) ${\textbf{splits are different than ours}}$; we ${\textbf{noticeably outperform}}$  GCN-LPA (Wang and Leskovec arxiv2020) and DFNet-ATT (Wijesinghe and Wang NeurIPS2019) on the same splits.
>
> ${\textbf{More detailed discussion}}$
> We have checked the papers on NodeNet and GCN-LPA (the GCN-LPA code is available, see the experiments and discussion below). The NodeNet paper does not provide the information about data splitting for Cora, CiteSeer, and PubMed datasets; the authors do not release their source code, and we have asked for code of NodeNet via email but got no reply yet. Importantly, based on Table 2 of NodeNet paper, we found that:
>
> (1) for Cora -- the authors compare the NodeNet with DFNet-ATT.
>
> (2) for Citeseer -- the authors compare NodeNet with GCN-LPA.
>
> In addition, we found that the results of baselines (i.e., DFNet-ATT and GCN-LPA) are copied from the original papers, i.e., DFNet-ATT (Wijesinghe and Wang NeurIPS2019) and GCN-LPA (Wang and Leskovec arxiv2020). Hence, we assume that the NodeNet follows the same data splits as DFNet-ATT (Wijesinghe and Wang NeurIPS2019) and GCN-LPA (Wang and Leskovec arxiv2020), and these ${\textbf{are different splits}}$ that we use in our paper. We choose the same dataset splits as in Yang et al. ICML2016 (which are also used in Kipf and Welling ICLR2017). The details of data splits are as follows: (1) Cora-ML: 140/500/1000; (2) CiteSeer: 120/500/1000; and (3) PubMed: 60/500/1000. However, GCN-LPA (Wang and Leskovec arxiv2020) split the training, validation, and test (i.e., Cora, CiteSeer, and PubMed) into 60%/20%/20%, which is totally different from our settings.
>
> Also, for DFNet-ATT (Wijesinghe and Wang NeurIPS2019), there was someone pointed out that “although you have the same training numbers (140 for cora) as Yang et al. [2016], but the labeled nodes in Yang is balanced, where there have 20 nodes for each class. But in your codes, the training nodes are selected from the first 140, and I verify that it's unbalanced.” (which can be found in Issues in DFNets Github) and the author created another version for DFNet on the correct/normal Cora dataset (i.e., the version we used in our experiment). Based on the corrected "load_data" function, we re-run DFNet-ATT on Cora and CiteSeer.
>
> For fair comparison, we perform (1) GCN-LPA (which follows the dataset splits in our paper) and (2) DFNet-ATT (where the labeled nodes in Cora-ML are balanced). The table below shows that our proposed TRI-GNN performs ${\textbf{significantly better}}$ than GCN-LPA and ${\textbf{better}$$ than DFNet-ATT. The mean accuracy (%) and standard deviation over 50 runs is reported for each dataset.
>
> ${^{\dagger}}$ means that we re-run GCN-LPA and DFNet-ATT based on the code provided by the authors.
>
> ++++++++++++++++++++++++++++++++++++++++++++
>
> Dataset | TRI-GNN$\quad$ | GCN-LPA $\quad$| DFNet-ATT
>
> ++++++++++++++++++++++++++++++++++++++++++++
>
> $\quad$ Cora | 84.98 (0.49) | ${^{\dagger}}$81.68 (0.93)  | ${^{\dagger}}$84.65 (0.50)
>
> ++++++++++++++++++++++++++++++++++++++++++++
>
> CiteSeer | 73.45 (0.65) | ${^{\dagger}}$71.40 (0.60)  |  ${^{\dagger}}$70.18 (0.71)
>
> ++++++++++++++++++++++++++++++++++++++++++++
>
> Once we get the code of NodeNet from the authors, we will add the results of NodeNet into our final version.

---

> ### Author Response · Authors · 2021-08-10
> **Response to Technical Question Regarding the Performance of NodeNet (Dabhi and Parmar arxiv2020) (1/2)**
>
> Q1: "For example, NodeNet achieves around 86% accuracy on node classification tasks on Cora."
>
> ${\bf A}$: ${\bf Updates}$: we just received the code today and try to get results later; also we have found the data split setting is indeed different from us and, hence, it is unfair to compare the published results from the NodeNet paper with our results.

---

> ### Author Response · Authors · 2021-08-18
> **Response to Technical Question Regarding the Performance of NodeNet (Dabhi and Parmar arxiv2020) (2/2)**
>
> Q1: "For example, NodeNet achieves around 86% accuracy on node classification tasks on Cora."
>
> ${\bf A}$: The pilot study below shows that our proposed TRI-GNN performs better than NodeNet on Cora, ACTIVSg200, and ACTIVSg500. Here, due to the specific dataset generation mode of NodeNet (i.e., in TFRecord format), in our comparative analysis with NodeNet (Dabhi and Parmar arxiv2020) on citation data, $\textbf{we follow the settings of NodeNet given by the NodeNet authors}$. That is, we split the Cora dataset into training (80%) and test sets (20%). For power grid networks (i.e., ACTIVSg200 and ACTIVSg500), we use a 10%/20%/70% split into training, validation, and test sets. We will add the full comparative analysis on all networks into the final version.
>
> ++++++++++++++++++++++++++++++++++++
>
> Dataset $\quad$ | TRI-GNN $\quad$ | NodeNet
>
> ++++++++++++++++++++++++++++++++++++
>
> Cora      $\qquad$ |    ${\bf 85.85 (0.71)}$       | 84.03 (0.40)
>
> ++++++++++++++++++++++++++++++++++++
>
> ACTIVSg200 | ${\bf 84.56 (1.75)}$ | 80.15 (0.91)
>
> ++++++++++++++++++++++++++++++++++++
>
> ACTIVSg500 | ${\bf 97.85 (0.56)}$ | 95.00 (0.50)
>
> ++++++++++++++++++++++++++++++++++++

---

### Official Review · Reviewer_oJSD · 2021-07-16

**Rating:** 7
**Confidence:** 3

**Summary:**

The work proposes an extension to conventional graph neural networks by
incorporating additional information about local topological structures in the
proximities of nodes.  The core contribution of the work is to additionally
allow rewiring of the graph based on the similarities of topological structures
of pairs of nodes.  It is shown that these rewirings can lead to increased
robustness of models against graph perturbations or adversarial attacks.  The
model is evaluated on a series of Node Classification tasks and is shown to
have favorable performance compared to many baselines.


**Limitations And Societal Impact:**

The authors have not addressed the social impact of their work.  While I think
that the implications are of limited degree, there are some potential
directions that could be mentioned here.  In particular, graph algorithms for
node classification can be used to categorize people using observational data
and include how they interact with others.  While this can
mostly be used for the good of developing better products or offering targeted
advertisement, it could also be used in surveillance systems where the social
implication would be non-negligible.

**Main Review:**

The work is well written and motivated and the paper is well structured.
Experiments are detailed and ablations are preformed to show the contribution
of individual model components (where some improvements could be made, see
major comment 1).  I have some concerns with regard to reproducability of the
results as hyperparameters are not exactly documented, further I am unable to
verify the selection process of hyperparameters (major comment 2).

Generally, the approach is novel and can represent a promising direction for
integrating aspects from topological data analysis into a Graph Neural
Networks.

Major comments:
 1. Removing individual components of the model can significantly impact the
    expressiveness of a model.  I would recommend to instead of removing the
    components, to implement fake versions which do not actually rely on PH.
    In the case of the TIMR ablation one could for example use random features
    as the basis of distance computations between nodes.  Without such
    ablations it is hard to tell if the reduced performance is due to lower
    model expressivity or due to the removal of the component.
 2. The search of hyperparameters seems to be largely limited to the presented
    method, which can give the approach a competitive advantage.  Further, the
    exact procedure for deriving the hyperparameters is not entirely clear.
    Which metric was used for deriving the best hyperparametes? Which ranges
    were evaluated?  Was the metric computed on a held out validation set or via
    cross validation?  What are the exact hyperparameters used for each model?
    The details are crucial to asses the study design and to estimate how
    representative the performance of the model is and pinpoint potential
    biases.

Minor comments:
 1. The graph corruption experiment could be extended to also incorporate
    corruption of node features.  While this is not covered by the stability
    theorem, it would still be interesting to see how the models perform based
    on different perturbations of the input.


**Time Spent Reviewing:**

4

---

> ### Author Response · Authors · 2021-08-06
> **Response to Societal Impact**
>
> We agree on the outlined societal impact. We also thought of a positive application of the proposed tool for matchmaking but cannot get the appropriate testing data at the moment.

---

> ### Author Response · Authors · 2021-08-06
> **Response to Technical Questions (1/2)**
>
> Q1: "Removing individual components of the model can significantly impact the expressiveness of a model. I would recommend to instead of removing the components, to implement fake versions which do not actually rely on PH. In the case of the TIMR ablation one could for example use random features as the basis of distance computations between nodes. Without such ablations it is hard to tell if the reduced performance is due to lower model expressivity or due to the removal of the component."
>
> ${\bf A}$: We agree with this suggestion, thank you. We are currently running experiments and will post our results soon.
>
>
> Q2: "The search of hyperparameters seems to be largely limited to the presented method, which can give the approach a competitive advantage. Further, the exact procedure for deriving the hyperparameters is not entirely clear. Which metric was used for deriving the best hyperparametes? Which ranges were evaluated? Was the metric computed on a held out validation set or via cross validation? What are the exact hyperparameters used for each model? The details are crucial to asses the study design and to estimate how representative the performance of the model is and pinpoint potential biases."
>
> ${\bf A}$: We use accuracy as the metric for deriving the best hyperparameters and the accuracy is computed via cross-validation. We fix and set the regularization weight to be $5e^{-4}$. For our model and baselines, we extensively tune hyperparameters by using grid search. We search the learning rate among {0.001, 0.005, 0.01, 0.05, 0.1}, hidden dimension among {8, 16, 32, 64, 128}, dropout rates within the range from 0.0 to 1.0 in increments of 0.1, and regularization parameter $\mu$ within the range from 0.1 to 1.0 in increments of 0.1. The optimal choice of eps1 and eps2 can be obtained via cross-validation. For instance, the optimal quantile of eps1 and eps2 for ACTIVSg500 dataset is 0.28 and 0.99 respectively. We consider the lower and upper bounds for eps1 are 0.01 and 2-quantile respectively and the lower and upper bounds for eps2 are 3-quantile and 0.99 respectively. For eps1, we generate a sequence from 0.01 to 0.50 with increment of the sequence 0.01; for eps2, we generate a sequence from 0.75 to 0.99 with increment of the sequence 0.01. Based on cross-validation, we conduct experiments in different eps1 and eps2 combinations. We have reported the training settings of our model in the supplementary material (Section C.2; line # 38 - 56).
>
> For baselines (on citation networks), we reuse the code and parameters (the reported best hyperparameter setting) released in the original paper. We will report the set of all hyperparameters in the final version. If the reviewer suggests, we can also report all hyperparameters for benchmark models in the rebuttal.
>
>
> Q3: "The graph corruption experiment could be extended to also incorporate corruption of node features. While this is not covered by the stability theorem, it would still be interesting to see how the models perform based on different perturbations of the input."
>
> ${\bf A}$: Thank you for this suggestion! It is a very interesting point we have been thinking about in terms of the new attack strategies. We are currently running experiments and will post our results soon.

---

> ### Author Response · Authors · 2021-08-09
> **Response to Technical Questions (2/2)**
>
> Q1: "Removing individual components of the model can significantly impact the expressiveness of a model. I would recommend to instead of removing the components, to implement fake versions which do not actually rely on PH. In the case of the TIMR ablation one could for example use random features as the basis of distance computations between nodes. Without such ablations it is hard to tell if the reduced performance is due to lower model expressivity or due to the removal of the component."
>
> ${\bf A}$: As suggested, we have conducted additional ablation experiments on our proposed TIMR based on random features. That is, for TRI-GNN$_{\text{A}}$ (node attributes as filtration function), we compute the Wasserstein distance among persistence diagrams which are generated based on random features (where the random features follow the uniform distribution $U(0,1)$). The table below shows that the performance based on topological signatures delivered by randomly generated features is significantly worse than the performance based on  topological signatures of the original node attributes on Cora-ML dataset. These ablation results indicate that the TRI-GNN$_\text{A}$ component of integrating topological information of the observed node features is critical.
>
> ++++++++++++++++++++++++++++++++++++++++++++++++++++++++++++++++++++
>
> Cora-ML  $\quad$  | TRI-GNN$_\text{A}$ with node attributes | TRI-GNN$_\text{A}$ with random features
>
> ++++++++++++++++++++++++++++++++++++++++++++++++++++++++++++++++++++
>
> Acc (%) (std) | $\quad$$\quad$$\quad$$\quad$84.80 (0.55) $\quad$$\quad$$\quad$ | 81.65 (0.90)
>
> ++++++++++++++++++++++++++++++++++++++++++++++++++++++++++++++++++++
>
>
> Q3: "The graph corruption experiment could be extended to also incorporate corruption of node features. While this is not covered by the stability theorem, it would still be interesting to see how the models perform based on different perturbations of the input."
>
> ${\bf A}$: We apply the state-of-the-art graph poisoning attack -- nettack (Z$\ddot{\text{u}}$gner et al. KDD2018) to perturb the Cora-ML dataset, i.e., adversarial perturbations targeting both the node’s features and the graph structure. From Table 1, we find that our proposed TRI-GNN model consistently outperforms other baselines under different perturbation rates. For instance, on Cora-ML dataset at 5 perturbation per targeted node, the improvement gain of TRI-GNN$_{\text{A}}$ over the next most accurate method is 9.65%. This result demonstrates that our model can also resist the targeted adversarial attack.
>
> We can post the results for mettack (Z$\ddot{\text{u}}$gner and G$\ddot{\text{u}}$nnemann ICLR2019) too.
>
> Table 1: Node classification performance (Accuracy $\pm$ Std) under nettack on Cora-ML.
>
> ++++++++++++++++++++++++++++++++++++++++++++++++++++++++++++++++++++++++
>
> \# of Perturbations Per Node | $\quad$ TRI-GNN$_{\text{A}}$ | $\quad$ GAT $\quad$ | $\quad$ RGCN$\quad$ | APPNP
>
> ++++++++++++++++++++++++++++++++++++++++++++++++++++++++++++++++++++++++
>
> $\quad$ $\quad$ $\quad$ $\quad$ $\quad$ 0 $\quad$$\quad$ $\quad$ $\quad$ | 84.80 (0.55) | 83.11 (0.70) | 82.80 (0.60) | 83.31 (0.53)
>
> ++++++++++++++++++++++++++++++++++++++++++++++++++++++++++++++++++++++++
>
> $\quad$$\quad$$\quad$ $\quad$$\quad$ $\quad$1$\quad$ $\quad$ $\quad$ $\quad$| 84.02 (0.45) | 77. 02 (0.74) | 77.29 (0.69) | 78. 25 (0.92)
>
> ++++++++++++++++++++++++++++++++++++++++++++++++++++++++++++++++++++++++
>
> $\quad$$\quad$$\quad$$\quad$ $\quad$$\quad$ 2$\quad$ $\quad$ $\quad$ $\quad$| 79.77 (0.80) | 74.00 (1.09) | 73.52 (0.87) | 75.23 (0.64)
>
> ++++++++++++++++++++++++++++++++++++++++++++++++++++++++++++++++++++++++
>
> $\quad$$\quad$$\quad$$\quad$ $\quad$$\quad$ 3 $\quad$$\quad$ $\quad$ $\quad$| 75.21 (0.71) | 66.26 (1.22) | 71.21 (0.96) |68.00 (1.20)
>
> ++++++++++++++++++++++++++++++++++++++++++++++++++++++++++++++++++++++++
>
> $\quad$$\quad$$\quad$$\quad$ $\quad$$\quad$ 4 $\quad$$\quad$ $\quad$ $\quad$| 70.25 (1.03) |63.56 (0.96) | 63.89 (1.40) |65.07 (1.13)
>
> ++++++++++++++++++++++++++++++++++++++++++++++++++++++++++++++++++++++++
>
> $\quad$$\quad$$\quad$$\quad$ $\quad$$\quad$ 5 $\quad$$\quad$ $\quad$ $\quad$| 68.39 (1.12) |61.79 (1.39) | 58.50 (1.47) | 62.11 (1.11)
>
> ++++++++++++++++++++++++++++++++++++++++++++++++++++++++++++++++++++++++

---

> > ### Comment · Reviewer_oJSD · 2021-09-02
> > **Thank you for the response and the additional experiments**
> >
> > Regarding the fake PH computation experiment, I actually meant an alternative approach where *no PH computation* is performed at all. I.e. that the persistence diagrams themselves are generated (at least partially randomly). Nevertheless, I think the experiment performed gives similar insights and thus addresses my concerns.
> >
> > Thank you for the additional graph poisoning experiments, these will make a good addition to the paper. On a side note, the better performance could also be potentially be explained by the discrete rewiring steps that the model implements as many attack and adversarial example approaches are based on gradients from the model (which due to the discrete rewiring are discontinuous).
> >
> > The additional experiments and clarifications address my concerns, yet I can also understand the concerns of the other reviewers.
> > Overall, I find that most points raised by the other reviewers are addressed quite well, such that I will improve my score by one point.

---

> > > ### Author Response · Authors · 2021-09-02
> > > **Feedback**
> > >
> > > We are very grateful for your positive feedback!
> > > Also, we have not thought before on the potential interpretation of TRI-GNN via its linkage to the gradient discontinuity. If it’s indeed true, then the TRI-GNN re-wiring strategy can be potentially optimized with respect to the type and number of discontinuities. Thank you very much for this idea!

---

### Official Review · Reviewer_srjf · 2021-07-20

**Rating:** 5
**Confidence:** 5

**Summary:**

The paper proposes the use of topological features into the graphs for the task of node classification. The proposed approach uses two modules for this purpose, the Topologically Induced Multigraph Representation module (TIMR) and the STAN (Subgraphs, Topology and Neighbour attribute) module. The TIMR module builds topological features from the k-hop neighbourhood of the graph. It finds the persistence diagrams (0 dimensional) of the k-hop neighborhood of every node. The filtration takes place depending on the node degree or the edge weights (computed by the distance between the node attributes). Once the persistence diagrams (PDs) for the nodes are found the pairwise Wasserstein distance is computed for these PDs. If it is less than a threshold (\epsilon1) a positive edge (+1) is added to the adjacency matrix and if it is greater than another threshold (\epsilon2) a negative edge (-1) is added. In the final adjacency matrix W^{topo} is defined as 1 if the combined sum in the previous step is greater than 0, else 0. This matrix is then used to find a modified Laplacian which is used in the GNN layer to aggregate node features from the STAN module. The STAN module is used to learn node features from the topology and node attributes. The edge weights are computed by normalizing over the Wasserstein distances in the neighbourhood of the node which is used in the weighted averaging followed by the aggregation layer that combines the neighbourhood information with the node’s features. The topology induced adjacency matrix (obtained from TIMR) and the node features (from the STAN module) are fed to the TRI-GNN layer with the proposed modified Laplacian and this module learns the parameters for performing the task of node classification. The proposed method performs well in comparison to the baselines.

**Ethics Review Area:**

["I don’t know"]

**Limitations And Societal Impact:**

There was no societal impact section in this paper.

**Main Review:**

-- Strengths:
1. Induces topological information from the node neighborhood by adding/deleting edges that are topologically similar/dissimilar

2. Learns node features from the edges weighted by the topological distance, thus introducing an attention over the neighbourhood based on topology

3. The proposed method performs well in comparison to the baselines

-- Weaknesses:
1. The paper states that the method uses 0 cycles for computing the PDs. These 0-dimensional PDs just contain connectivity information in terms of the number of connected components, which can easily be deduced with other graph-theoretic algorithms. Is there a reason why authors do not use any higher-dimensional PDs? Also the paper states in section 4.1 regarding the aim “Our aim is to systematically extract all k-dimensional topological features and their persistence in each node neighborhood and then to compare node neighborhoods in terms of their exhibited shapes“. However this doesn’t seem to be backed up empirically.

2. A thorough time-complexity analysis would have been more useful. The overall time-complexity of the method seems quite large considering that PDs have to be computed per node in addition to pairwise Wasserstein distances. It looks like the method would not scale well to large (or massive) graphs. Please address this point in the rebuttal.

3. The added topological information has not been learned end-to-end in a differentiable manner thus preventing any adaptive learning. For example, if the existence of a homology class in the neighbourhood of a node would result in that node belonging to a certain class. Ideally this structure should be detected and used for prediction. However this would not be possible by simply using the hardcoded topological features.

4. This work seems to leverage homophily as an inductive bias. I am not sure why edges must be removed between nodes whose neighborhood topology differs? This idea might not work so well for graphs which exhibit a lot of heterogeneity. For example, a graph with bank customers as nodes and some transaction history dictating edges (and their weights) between customers, can easily have forgers or attackers as nodes in the graph, that are surrounded with customer nodes (that are very different from the attacker). Such a situation is quite realistic in real-life.

5. The graphs picked for the experiments might be exhibiting a high degree of homophily among nodes. It would be interesting to see how this method performs on non-homophilous datasets. Such datasets can be found in:
“New Benchmarks for Learning on Non-Homophilous Graphs” - Lim et. al (WWW’21).

6. In Theorem 1, K(eps1,eps2) is a function dependent on eps1 and eps2. I am not sure how this bound is of any use because K(eps1,eps2) could be arbitrarily large or even unbounded depending on the choice of eps1 and eps2. It would be more useful to present a study about how these bounds behave in different regimes and choices of parameters eps1 and eps2. How are eps1 and eps2 determined? This is not discussed in the paper.

7. In definition 4.1, how is the “k” in the k-hop neighborhood determined? How does it work in the case of sparse vs. dense graphs? Or graphs with low average degrees vs. those with high-average degrees?

8. It would be more interesting to use graph adversarial attacks in the experiments that just don't perturb random edges/nodes etc. but do something smarter. E.g.
"Towards More Practical Adversarial Attacks on Graph Neural Networks" - Ma. et. al. (NeurIPS 20).


**Needs Ethics Review:**

Yes

**Time Spent Reviewing:**

3-4 hours

---

> ### Author Response · Authors · 2021-08-06
> **Response to Technical Questions (2/2: Q5, Q6, Q7, Q8)**
>
> Q5: "The graphs picked for the experiments might be exhibiting a high degree of homophily among nodes. It would be interesting to see how this method performs on non-homophilous datasets. Such datasets can be found in: “New Benchmarks for Learning on Non-Homophilous Graphs” - Lim et. al (WWW’21)."
>
> ${\bf A}$: We thank the reviewer for noting that the non-homophilous datasets could be extended. Based on the paper Lim et. al (WWW’21), we have run our proposed model on deezer-europe dataset. Table 1 depicts the results for node classification on a non-homophilous dataset - deezer-europe and baselines include three top-performance models and PEGN-RC; we observe that our proposed TRI-GNN\textsubscript{D} model outperforms state-of-the-art baselines, which implies that our TRI-GNN model is able to provide reasonable power and perform well in non-homophilous setting. The improvement gain of TRI-GNN$_{\text{D}}$ over the baselines ranges from 4.11% to 4.56%.
>
> Table 2: Classification accuracies (standard deviation) on deezer-europe
> (non-homophilous) dataset.
> ++++++++++++++++++++++++++++++++++++++++++++++++++++++++++++++++
>
> Dataset | TRI-GNN$_{\text{D}}$ |    $\enspace$  $\enspace$ APPNP $\enspace$   |    $\enspace$ H2GCN    $\enspace$  |    GPR-GNN $\enspace$  |   PEGN-RC
>
> ++++++++++++++++++++++++++++++++++++++++++++++++++++++++++++++++
>
> deezer   |        70.10 (0.79)         |   67.21 (0.56) |  67.22 (0.90) |   66.90 (0.50)  | 67.13 (1.21)
>
> ++++++++++++++++++++++++++++++++++++++++++++++++++++++++++++++++
>
>
> Q6: "In Theorem 1, K(eps1,eps2) is a function dependent on eps1 and eps2. I am not sure how this bound is of any use because K(eps1,eps2) could be arbitrarily large or even unbounded depending on the choice of eps1 and eps2. It would be more useful to present a study about how these bounds behave in different regimes and choices of parameters eps1 and eps2. How are eps1 and eps2 determined? This is not discussed in the paper."
>
> ${\bf A}$: Yes, K can be very large depending on eps1 and eps2, but our point is that if the original graphs $G^+$ and $G^-$ are topologically similar, then this implies the average degrees of the rewired graphs $T^+$ and $T^-$ are to be close to each other. If these average degrees are far from each other, then $G^+$ and $G^-$ are topologically very different from each other, too. That is, the topological similarity of the original graphs controls in some way the average degrees of the induced rewired graphs.
>
> In practice, we select the eps1 and eps2 values based on quantiles of Wasserstein distances between persistence diagrams. The optimal choice of eps1 and eps2 can be obtained via cross-validation. For instance, the optimal quantile of eps1 and eps2 for ACTIVSg500 dataset is 0.28 and 0.99 respectively. We consider the lower and upper bounds for eps1 are 0.01 and 2-quantile, respectively, and the lower and upper bounds for eps2 are 3-quantile and 0.99, respectively. For eps1, we generate a sequence from 0.01 to 0.50 with increment of the sequence 0.01; for eps2, we generate a sequence from 0.75 to 0.99 with increment of the sequence 0.01. Based on cross-validation, we conduct experiments in different eps1 and eps2 combinations. Tables 3 and 4 show the performances with respect to different thresholds (eps1 and eps2) combinations.
>
> We find that (eps1, eps2) around (0.30, 0.99) tend to deliver the optimal performance. The optimal results differ among graphs and depend on sparsity, label rates and graph higher-order properties.
>
> Table 3: Results of TRI-GNN with different thresholds (eps1 and fixed eps2) selections.
> +++++++++++++++++++++++++++++++++++++++++++++++++++++++++++++++++++++
>
> (eps1, eps2) | (0.28, 0.99) |  (0.10, 0.95) | (0.20, 0.95)  | (0.30, 0.95)  | (0.40, 0.95)
>
> +++++++++++++++++++++++++++++++++++++++++++++++++++++++++++++++++++++
>
> ACTIVSg500 | 98.11 (0.43)| 94.57 (0.47) | 94.80 (0.77) | 95.85 (0.61) | 94.17 (0.68)
>
> +++++++++++++++++++++++++++++++++++++++++++++++++++++++++++++++++++++
>
>
> Table 4: Results of TRI-GNN with different thresholds (fixed eps1 and eps2) selections.
> +++++++++++++++++++++++++++++++++++++++++++++++++++++++++++++++++++++
>
> (eps1, eps2) | (0.30, 0.95) |  (0.30, 0.96) | (0.30, 0.97)  | (0.30, 0.98)  | (0.30, 0.99)
>
> +++++++++++++++++++++++++++++++++++++++++++++++++++++++++++++++++++++
>
> ACTIVSg500 | 95.85 (0.61)| 95.86 (0.71) | 96.70 (0.32) | 97.80 (0.55) | 97.89 (0.34)
> +++++++++++++++++++++++++++++++++++++++++++++++++++++++++++++++++++++
>
> Q7: "In definition 4.1, how is the “k” in the k-hop neighborhood determined? How does it work in the case of sparse vs. dense graphs? Or graphs with low average degrees vs. those with high-average degrees?"
>
> ${\bf A}$: The choice of $k$ is balanced by two goals: to collect sufficiently rich topological information from each node neighborhood and to maintain computational costs at a lower level. In our analysis we use cross validation, and as expected, if the average degree of the ambient graph is low, then we tend to choose larger $k$, while if the average degree is high, we choose $k$ small.
>
> In addition, for graphs with lower average degree, we can consider two other routes which are closely related to the choice of $k$. The first alternative is to use attribute based filtration which is not impacted by graph sparsity. The second approach we suggest is power filtration. In sparse graphs, power filtration is an effective tool to distinguish between similar nodes. It is computationally costly in dense graphs, but efficient in sparse graphs.
>
>
> Q8: "It would be more interesting to use graph adversarial attacks in the experiments that just don't perturb random edges/nodes etc. but do something smarter. E.g. "Towards More Practical Adversarial Attacks on Graph Neural Networks" - Ma. et. al. (NeurIPS 20)."
>
> ${\bf A}$: Thank you for suggesting this paper which we have been unfamiliar with. Based on the paper “Towards More Practical Adversarial Attacks on Graph Neural Networks” - Ma. et. al. (NeurIPS 20), we conduct two kinds of experiments: degree attack and RWCS. Tables 5 and 6 show the accuracies (%) and standard deviation in () of our proposed TRI-GNN$_{\text{D}}$ and baseline methods. Tables 5 and 6 demonstrate that our proposed TRI-GNN model is more robust against more practical/smarter attacks compared with baseline methods, especially under RWCS attack.
>
> Table 5: Comparison of the accuracy of TRI-GNN$_{\text{D}}$ and baseline methods under degree attack.
>
> ++++++++++++++++++++++++++++++++++++++++++++++++++++++++++++
>
> Cora-ML  | TRI-GNN$_{\text{D}}$ |       $\enspace$  GCN  $\enspace$  | JKNetConcat |     APPNP   |    RGCN
>
> ++++++++++++++++++++++++++++++++++++++++++++++++++++++++++++
>
> Acc. (std) |    82.0 (0.9)    | 78.2 (0.4) |    $\enspace$ 60.7 (1.0)  $\enspace$  | 80.0 (0.3) |   81.5 (1.2)
>
> ++++++++++++++++++++++++++++++++++++++++++++++++++++++++++++
>
> Table 6: Comparison of the accuracy of TRI-GNN$_{\text{D}}$ and baseline methods under RWCS attack.
>
> ++++++++++++++++++++++++++++++++++++++++++++++++++++++++++++
>
> Cora-ML  | TRI-GNN$_{\text{D}}$ |    $\enspace$  GCN  $\enspace$   | JKNetConcat |     APPNP   |    RGCN
>
> ++++++++++++++++++++++++++++++++++++++++++++++++++++++++++++
>
> Acc. (std) |    82.6 (0.7)    | 79.5 (0.3) |    $\enspace$ 71.2 (0.5) $\enspace$   | 81.5 (0.9) |   78.2 (0.3)
>
> ++++++++++++++++++++++++++++++++++++++++++++++++++++++++++++

---

> ### Author Response · Authors · 2021-08-06
> **Response to Technical Questions (1/2: Q1, Q2, Q3, Q4)**
>
> Q1: "The paper states that the method uses 0 cycles for computing the PDs. These 0-dimensional PDs just contain connectivity information in terms of the number of connected components, which can easily be deduced with other graph-theoretic algorithms. Is there a reason why authors do not use any higher-dimensional PDs?  Also the paper states in section 4.1 regarding the aim “Our aim is to systematically extract all k-dimensional topological features and their persistence in each node neighborhood and then to compare node neighborhoods in terms of their exhibited shapes“. However this doesn’t seem to be backed up empirically."
>
> ${\bf A}$:  Thank you for this question, as it interestingly connects to a number of fundamental problems with TDA on graphs. The primary reason we consider 0 cycles is computational complexity vs. classification accuracy. We have found that performance gains due to including 1-dimensional features are negligible (see Table 1 below for a case study on Cora-ML and results for other graphs are similar). However, the method, in general, can handle any k-dimensional features and in some data, e.g., financial transaction graphs, higher order cycles are often more important.
>
> Table 1. Accuracy (%) (standard deviation in ()) when comparing TRI-GNN${_\text{D}}$ with 0-dimensional feature with ${_\text{D}}$ with 0- and 1-dimensional features on Cora-ML dataset.
>
> ++++++++++++++++++++++++++++++++++++++++++++++++++++++++++++++++++++++++++++++++++++++++++
>
> Cora-ML | TRI-GNN${_\text{D}}$ with 0-dimensional feature | TRI-GNN${_\text{D}}$ with 0- and 1-dimensional features
> ++++++++++++++++++++++++++++++++++++++++++++++++++++++++++++++++++++++++++++++++++++++++++
>
> Acc (std)|                     84.98 (0.49)                 $\quad \quad \quad \quad \quad \quad \quad \quad \quad \quad$                             |                         84.11 (0.72)
> ++++++++++++++++++++++++++++++++++++++++++++++++++++++++++++++++++++++++++++++++++++++++++
>
> Furthermore, the reviewer is right that 0-dimensional persistent homology is about keeping track of the connected components, however there is no graph theoretic algorithm to capture such topological information. The difference of persistent homology (PH) here is that it takes a filtration (a sequence of subspaces) and records the interrelation of these subspaces. For example, this filtering positions the graph with respect to a function - in our case, the degree function, where the lowest degree nodes are at the bottom and the highest degree nodes are at the top. Then, in spirit of the mapper technique, 0-dimensional persistent homology captures the coarse shape of the graph with respect to this filtering function just like the mapper algorithm induces a simple (Reeb) graph of the topological space, which is basically the “coarse shape” of the space. Here, the key is the choice of the filtering function. In our case, we tried several functions in different datasets. Since we took fixed radius neighborhoods of the nodes (i.e., $k$), the degree function worked best among the graph inherent functions. This is because the other distance (position) based functions (e.g., centrality and eccentricity) are very limited to differentiate among these fixed radius neighborhoods since in these $k$-hop neighborhoods ${G_{u}^k}$ these functions tend to deliver very similar values for each node as their maximum can be the diameter ($2k$) for small $k$. This makes the filtration sequence very short, and persistent homology does not work well to distinguish different topological structures.
>
> Finally, for each node in a graph, we took its $k$-hop neighborhoods and compared the topologies of these neighborhoods with respect to the degree function as the filtering function. Since we used the same filtering function for each $k$-hop neighborhood, it gave us a fair comparison for the similarities of their coarse shapes.
>
> In the paper, we also used attribute-induced PH where the edge weights are induced from the node attributes. In general, for sparser heterogeneous graphs with lower numbers of node attributes per class, richer topological structure and weaker dependency between attributes and classes, PH induced by the graph properties works better than attribute-induced PH, while for denser, more homogeneous graphs as IEEE118 and Citeseer, attribute-based filtration is the preferred choice. However, even in case of attribute-induced PH, we find minimal gains due to including 1-dimensional cycles.
>
> Q2: "A thorough time-complexity analysis would have been more useful. The overall time-complexity of the method seems quite large considering that PDs have to be computed per node in addition to pairwise Wasserstein distances. It looks like the method would not scale well to large (or massive) graphs. Please address this point in the rebuttal."
>
> ${\bf A}$: Indeed, scalability can be an issue, as in most PH studies on graphs. Currently, this is the primary bottleneck problem for TDA on graphs. However, running time of our TRI-GNN and its competitor PEGN-RC of Zhao et al. is comparable, e.g., for power grid network: ACTIVSg500  139.77 sec (TRI-GNN) vs. 105.09 sec (PEGN-RC) and for citation network: Cora-ML 211.78 sec (TRI-GNN) vs. 208.20 sec (PEGN-RC) (results for other graphs are in the supplementary material Section C.5, Table IV).
>
> Also, to reduce computational complexity for very dense graphs, we may use a degree filtration which is still found to be very powerful even for smaller $k$-hops. Further reduction in computational costs can be done using landmarks (Silva and Carlsson, Eurographics 2004) and various approximate simplicial representations.
>
>
> Q3: "The added topological information has not been learned end-to-end in a differentiable manner thus preventing any adaptive learning. For example, if the existence of a homology class in the neighborhood of a node would result in that node belonging to a certain class. Ideally this structure should be detected and used for prediction. However this would not be possible by simply using the hardcoded topological features."
>
> ${\bf A}$:  We agree with this comment. In case of graph classification (as opposed to node classification considered here), the common tool for end-to-end learning topological information in a differentiable manner is either kernelization of PDs or persistence images (PIs) (e.g., Hofer et al. NeurIPS2017, Kusano​​ et al. ICML2016). This approach can be extended to our case, with similarity among neighborhoods defined, e.g., via the distance in the corresponding Reproducing kernel Hilbert spaces (RKHS). In our pilot results on clusterization and classification of nodes using their local topological info, so far we have found no substantial gains due using such a learning scheme. However, we agree that with a further extended GNN architecture, it is a very promising direction.
>
>
> Q4: "This work seems to leverage homophily as an inductive bias. I am not sure why edges must be removed between nodes whose neighborhood topology differs? This idea might not work so well for graphs which exhibit a lot of heterogeneity. For example, a graph with bank customers as nodes and some transaction history dictating edges (and their weights) between customers, can easily have forgers or attackers as nodes in the graph, that are surrounded with customer nodes (that are very different from the attacker). Such a situation is quite realistic in real-life."
>
> ${\bf A}$: Actually, the bank customer example may be a good illustration why our approach works! Our learning mechanism accounts for both interactions of the node (i.e., the attacker) with its neighbors (i.e., good customers) and interactions among the neighbors (i.e., good customers). As it is harder for the attacker to establish exactly the same topology of mutual interactions with good customers, as other clean nodes do, the resulting PD of the attacker will be different than PDs of the clean nodes. Removing an edge from such an attacked node reduces the impact of the attack on classification accuracy. More generally, while the idea is certainly rooted in homophily, we propose here a “topological homophily”, i.e. similarity of neighborhood shapes rather than individual node features. In our experiments we consider sparse heterogeneous networks with density from 0.0002 to 0.0009 and find that TRI-GNN performance gains in these networks under attacks range from 0.74% to 4.30%.

---

### Review · Ethics_Reviewer_7ABz · 2021-08-12

**Recommendation:**

I believe the authors can at least flag concerns about the use of the approach through an exploration of the dating example they use in the opening.

**Ethical Issues:**

Yes

**Ethics Review:**

The authors explain the benefit of their work via this example:

"As a result, our new topology-induced multigraph representation (TIMR) of graph G in (i) strengthens the relationship among nodes which might not be (yet) connected by a visible (or "tangible") edge in G but whose higher order intrinsic characteristics are very similar. The intuitive analogy here might be with a matchmaking agency who aims to connect a male and a female on a blind date, based on a careful assessment of interplay among similarities of their interests, socio-demographics as well as that of their close friends, rather than bringing in two individuals together through a random walk among all available profiles of males and females. Second, such a new multiedge in (i), based on shape similarity of node local neighborhoods may assist not only the matchmaking agency in coupling the most appropriate individuals (or link prediction in less romantic and more technical terms), but to enhance resilience of the graph structure and associated graph learning."

The example suggests several ethical considerations:
What higher order intrinsic characteristics are being identified? Given the example chosen might these be attributes of individuals or communities that they have intentionally withheld?
Could the "intrinsic characteristics" infer relationships or similarities among groups that the groups would consider wrong or objectionable?
How does this technique in domains such as dating or social media relate to/contribute to concerns with "filter bubbles" and "echo chambers"?

Given that you lead with an example in a social domain, dating, it would be useful to think more fully about what it means to infer higher order intrinsic characteristics and how such inferences might both leak private (or at least withheld) information (remember the gaydar paper) or reinforce polarization.

As another reviewer noted graph algorithms for node classification can be used in a wide range of domains where such inferences might raise social, ethical, and political concerns including policing, advertising, and politics.

Smaller point, there is no need to specify the genders of the potential daters.

---

> ### Author Response · Authors · 2021-08-15
> **Response to the Questions from Ethics Reviewer**
>
> Q: ``The example suggests several ethical considerations: What higher order intrinsic characteristics are being identified? Given the example chosen might these be attributes of individuals or communities that they have intentionally withheld? Could the "intrinsic characteristics" infer relationships or similarities among groups that the groups would consider wrong or objectionable? How does this technique in domains such as dating or social media relate to/contribute to concerns with "filter bubbles" and "echo chambers"?"
>
> ${\bf A}$: Thank you for some interesting food for thought! When we talk on “higher order intrinsic characteristics”, we mean higher-order interactions among nodes and their attributes captured by the analysis of simplicial complexes of varying orders. Now, the problem of intentionally withholding/manipulating some information is linked to the problem of creating a new node attribute interaction graph whose persistent homology matches the desired goal, i.e., design of a graph system with a pre-defined simplicial complex representations (potentially for any order $k$ of simplicial complex). It shall be NP hard. Though, in reality we do not look at all orders of simplicial complexes and some manipulation is possible but it hardly will be fruitful (to address it, we can look at multiple filtrations and compare results; each filtration will results in its own sequence of simplicial complexes that manipulators need to match).
>
> The problem of  "filter bubbles" and "echo chambers" may indeed affect the outcome of our method and again is linked to the bias/fairness in AI issues, especially if the data from echo chambers substantially prevail. In case of the online dating example, it may not be a very negative outcome (at least, it is debatable) as both candidates are likely to be chosen from the same chambers and as such are likelier to share many traits and form a good personal match. However, in other applications, e.g., various surveys on social networks, biosurveillance using online social media, the implications will be highly negative and ought to be addressed.
>
> Mitigation of such bias is a problem on its own but the primary goal is to make node neighborhoods more diverse: some ideas include using multiple filtration functions based on node attributes so the node neighborhoods are made more diverse and are no longer dominated by a specific group; remove some node attributes completely from the study (which has its own pros and cons in terms of accuracy), and create synthetic topological signatures using bootstrap zigzag quivers to mitigate the data bias and its effects.
>
> Q: ``Given that you lead with an example in a social domain, dating, it would be useful to think more fully about what it means to infer higher order intrinsic characteristics and how such inferences might both leak private (or at least withheld) information (remember the gaydar paper) or reinforce polarization.''
>
> ${\bf A}$: We agree this is a very good point. Topological methods are not immune to potentially revealing some sensitive information. We will think on the intersection of privacy and topological ML (this is really a super-novel area) and will add our thoughts into the final version.

---

### Review · Ethics_Reviewer_73oe · 2021-08-13

**Recommendation:**

A societal impact statement can be added to an updated version of this paper, which would address the key concern here.

**Ethics Review:**

This paper proposes an approach to tackle smoothing problems in GNNs by integrating higher-order graph information. In reading this paper, I saw no overt ethical issues that the authors expressly overlook. The principal concern is that the authors do not include a societal impact statement.

---

> ### Author Response · Authors · 2021-08-15
> **Response to the Questions from Ethics Reviewer**
>
> Recommendation: ``A societal impact statement can be added to an updated version of this paper, which would address the key concern here.''
>
> ${\bf A}$: Yes, we will add the societal impact into our paper and the limitations section. Some of the questions raised by the reviewers were extremely helpful for thinking in this direction (thank you!). Among the major limitations we currently see is probable existence of the bias of the proposed node classification approach due to potentially unbalanced population groups, and the associated fairness in AI implications. We have not explored this bias before (and have not thought on such fairness aspects till the reviewers asked on the potential negative social implications) and there aldi currently exist no results on bias and fairness in topological methods in machine learning (ML). However, in the context of social data analysis, given that we analyze peer neighborhoods which themselves tend to be largely formed by the same users as the target node and hence are likely to be dominated by the abundant group, such bias is very likely to exist and to be non-negligible. Mitigating it is a problem on its own but some ideas include using multiple filtration functions based on different node attributes so the node neighborhoods are made more diverse and are no longer dominated by a specific group; we can also remove some potentially discriminative node attributes completely from the study (which has its own pros and cons in terms of accuracy), and we can also create synthetic topological signatures using bootstrap zigzag quivers to mitigate the data bias and its fairness implications. Another important limitation (inspired by the Reviewers comments) is the unexplored interaction of data privacy and topological methods in ML, particularly, in the context of revealing sensitive information.

---

### Decision · Program_Chairs · 2021-09-27

**Decision:**

Accept (Poster)

**Comment:**

While the paper initially obtained quite mixed reviews and some reviewers flagged ethical concerns, a substantial amount of constructive discussion among the authors and the expert reviewers (both technical and ethical) eventually clarified many issues. Taking into account all changes and clarifications mentioned (and promised) by the authors, the revised manuscript will be a valuable contribution to the field. I do, however, encourage the authors to take all comments seriously and adjust the manuscript accordingly (especially regarding notation, presentation, choice of hyperparameters, social impact, and an honest discussion of limitations).